# Gen-DFL: Decision-Focused Generative Learning for Robust Decision Making

**Prince Zizhuang Wang**
Heinz College
Carnegie Mellon University
`princewang@cmu.edu`

**Shuyi Chen**
Heinz College
Carnegie Mellon University
`shuyic@andrew.cmu.edu`

**Jinhao Liang**
Department of Computer Science
University of Virginia
`njs4nu@virginia.edu`

**Ferdinando Fioretto**
Department of Computer Science
University of Virginia
`spq7wp@virginia.edu`

**Shixiang Zhu**
Heinz College
Carnegie Mellon University
`shixiangzhu@cmu.edu`

## Abstract

Decision-focused learning (DFL) integrates predictive models with downstream optimization, directly training machine learning models to minimize decision errors. While DFL has been shown to provide substantial advantages when compared to a counterpart that treats the predictive and prescriptive models separately, it has also been shown to struggle in high-dimensional and risk-sensitive settings, limiting its applicability in real-world settings. To address this limitation, this paper introduces decision-focused generative learning (`Gen-DFL`), a novel framework that leverages generative models to adaptively model uncertainty and improve decision quality. Instead of relying on fixed uncertainty sets, `Gen-DFL` learns a structured representation of the optimization parameters and samples from the tail regions of the learned distribution to enhance robustness against worst-case scenarios. This approach mitigates over-conservatism while capturing complex dependencies in the parameter space. The paper shows, theoretically, that `Gen-DFL` achieves improved worst-case performance bounds compared to traditional DFL. Empirically, we evaluate `Gen-DFL` on various scheduling and logistics problems, demonstrating its strong performance against existing DFL methods.

## 1 Introduction

Decision-making under uncertainty arises in many real-world applications, including supply chain management, energy grid optimization, portfolio management, and transportation planning Sahinidis (2004); Liu & Liu (2009); Garlappi et al. (2006); Delage & Ye (2010); Hu et al. (2016); Kim et al. (2005); Chen et al. (2025b). In these settings, decision-makers act with incomplete information and use machine learning predictions to estimate uncertain parameters, such as future demand, outage risk in power grids, asset returns in portfolios, and travel times or flows in transportation systems.

Standard methods, commonly referred to as predict-then-optimize (PTO) Elmachtoub & Grigas (2017), tackle this problem by first training a predictive model to estimate the parameters of an optimization problem (*e.g.*, expected demand or cost coefficients) and then using these estimates as inputs to an optimization model. While the separation between prediction and optimization enhances efficiency, it also introduces a fundamental drawback. Predictive models are typically trained to minimize standard loss functions (*e.g.*, mean squared error), which may not align with the true objective of minimizing decision costs. As a result, small prediction errors can propagate through the optimization process, leading to costly, suboptimal decisions. For instance, in power outage

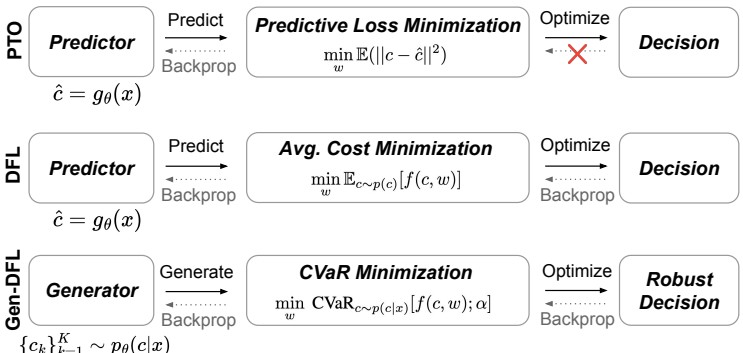

Figure 1: Comparison of the proposed decision-focused generative learning Gen-DFL ramework with conventional predict-then-optimize (PTO) and decision-focused learning (DFL).

management Zhu et al. (2021), overestimating energy demand may lead to unnecessary resource allocation, whereas underestimation could result in supply shortages and prolonged downtime.

To address this issue, decision-focused learning (DFL) integrates prediction and optimization into a single end-to-end framework Donti et al. (2017); Mandi et al. (2024b). Instead of optimizing purely for predictive accuracy, DFL trains machine learning models with the explicit goal of minimizing the final decision cost. This key idea is enabled by differentiating the optimization process within the learning loop, and results in an alignment of the model's predictions with their downstream impact. This approach has shown clear improvements in structured decision-making tasks where the optimization landscape is well-behaved and relatively low-dimensional.
Despite these advantages, DFL suffers from several critical limitations: ($i$) *Scalability*: In high-dimensional settings, the curse of dimensionality Köppen (2000) degrades the predictive model's ability to capture complex dependencies in the parameter space. Since DFL typically relies on single-point predictions, it struggles to encode the full distributional uncertainty of the decision variables Mandi et al. (2024a). This leads to overconfident estimates that degrade decision quality when uncertainty is high. ($ii$) *Risk Sensitivity*: In many applications, decision-makers prioritize robustness over worst-case outcomes rather than optimizing for expected performance. Traditional DFL models, however, are primarily trained to improve average-case decisions and do not explicitly model tail risks Ben-Tal et al. (2009); Beyer & Sendhoff (2007).

To overcome these challenges, this paper proposes decision-focused generative learning (Gen-DFL), a novel end-to-end framework that leverages generative models to enhance decision quality in high-dimensional and risk-sensitive settings. Unlike traditional approaches that rely on fixed uncertainty sets, Gen-DFL learns a distributional representation of uncertain parameters using deep generative models. Recent advances in generative modeling enable efficient learning of complex, high-dimensional distributions Dong et al. (2023); Wu et al. (2024), allowing for adaptive sampling from tail regions to support risk-aware decision-making without excessive conservatism. By dynamically balancing robustness and efficiency, Gen-DFL provides a more flexible and principled approach to decision optimization. A schematic comparison of the predict-then-optimize (PTO) model, standard DFL, and Gen-DFL is shown in Figure 1.

**Contributions.** The paper makes three contributions:

- It introduces Gen-DFL, a DFL framework that leverages generative models to capture uncertainty in high-dimensional stochastic optimization and to enable task-specific robustness control.
- It provides a theoretical analysis that characterizes conditions under which Gen-DFL outperforms traditional DFL, with emphasis on high-dimensional and risk-sensitive decision problems.
- Through experiments on both synthetic and real-world decision-making tasks, it shows that Gen-DFL improves decision quality compared to existing DFL baselines.

## 2 RELATED WORKS

Decision-focused learning (DFL) enhances decision-making under uncertainty by integrating prediction and optimization into a single framework. Bengio (1997) showed that optimizing predictive

models for decision outcomes improves financial performance. Differentiable optimization layers have further expanded DFL applications Agrawal et al. (2019). For example, Amos & Kolter (2017) introduced differentiable quadratic programs, enabling backpropagation through constrained optimization, while Agrawal et al. (2019) extended this to all convex programs. Parallel work has explored integrating integer programming into neural networks Mandi & Guns (2020); Wilder et al. (2019). There is also another line of research which focuses on improving efficiency and effectiveness of prediction-based DFL Shah et al. (2022; 2024); Kong et al. (2022).

However, existing DFL methods rely on single-point predictions, failing to capture uncertainty and leading to suboptimal decisions Köppen (2000); Ben-Tal et al. (2009). Additionally, they typically optimize for average-case performance, making them unsuitable for risk-sensitive applications Mandi et al. (2024b) in safety-critical or regulated domains. Approaches like Conformal-Predict-Then-Optimize (CPO) Patel et al. (2024) attempt to address this by constructing fixed uncertainty sets but can be overly conservative, especially in high-dimensional settings.

Robust Optimization (RO) provides a principled approach to decision-making under uncertainty by ensuring solutions remain feasible under the worst-case scenario Ben-Tal & Nemirovski (2002); Bertsimas & Thiele (2004); Ben-Tal et al. (2006), with clear guarantees and scalability. Instead of relying on probabilistic assumptions about uncertain parameters, RO constructs uncertainty sets that define the range of possible parameter values Bertsimas et al. (2011) and aims to find the decision that is robust against the worst-case in the uncertainty sets. This approach has found applications in domains such as supply chains Bertsimas & Thiele (2004), currency portfolio management Fonseca et al. (2011), and power system optimization Liang et al. (2024).

Despite its guarantees, the solutions suggested by RO suffer from two major limitations: ($i$) Uncertainty set construction usually relies on heuristic choices, making it difficult to capture the real dynamics in the real-world applications Liang et al. (2024). ($ii$) Such pre-specified uncertainty sets tend to be overly conservative Roos & den Hertog (2020) as it focuses solely on the worst-case outcome Wang et al. (2025); Chenreddy & Delage (2024); Yeh et al. (2024), whereas many high-stakes applications require accounting for multiple adverse scenarios.

## 3 PRELIMINARIES

**Decision-Focused Learning.** Consider a general stochastic optimization problem:

$$w^\star := \arg\min_w \mathbb{E}_{c \sim p(c)}[f(c, w)], \tag{1}$$

where $c$ is a random vector characterizing the problem parameters and $f(c, w)$ is the objective function. The goal is to find the optimal decision $w^\star$ that minimizes the expected decision cost under the distribution $p(c)$, given the true data-generating process.

A common approach, predict-then-optimize (PTO), assumes a linear objective, which simplifies the problem to the following deterministic surrogate:

$$w^*(\hat{c}) := \arg\min_w \hat{c}^T w, \tag{2}$$

where $\hat{c}$ is the estimate of $\mathbb{E}[c|x]$ conditioning on covariate $x$. This framework consists of two components: ($i$) A predictor $\hat{c} := g_\theta(x)$, trained to minimize the standard mean squared error (MSE) $\mathbb{E}||\hat{c} - c||^2$; ($ii$) An optimization model that finds the best decision $w$ given $\hat{c}$. As noted by Elmachtoub & Grigas (2017), this approach often leads to suboptimal decisions, as minimizing prediction error does not necessarily translate to improved decision quality.

To mitigate this issue, decision-focused learning (DFL) Mandi et al. (2024b) integrates prediction with decision-making by training $g_\theta(x)$ using decision regret as the loss function. The resulting objective takes the following form:

$$\ell_{\text{DFL}}(\theta) = \mathbb{E}_x \left[ \text{Regret}(g_\theta(x), c) \right], \text{ where } \text{Regret}(g_\theta(x), c) = f(c, w^\star(g_\theta(x))) - f(c, w^\star(c)).$$

For notational simplicity, we use $c$ to denote the true mean of the optimization parameters given $x$. By optimizing $g_\theta(x)$ directly with respect to decision performance, DFL ensures that the predicted parameters yield decisions that are robust to downstream cost objectives. We will refer to this conventional DFL approach, which relies on explicit prediction models, as Pred-DFL.

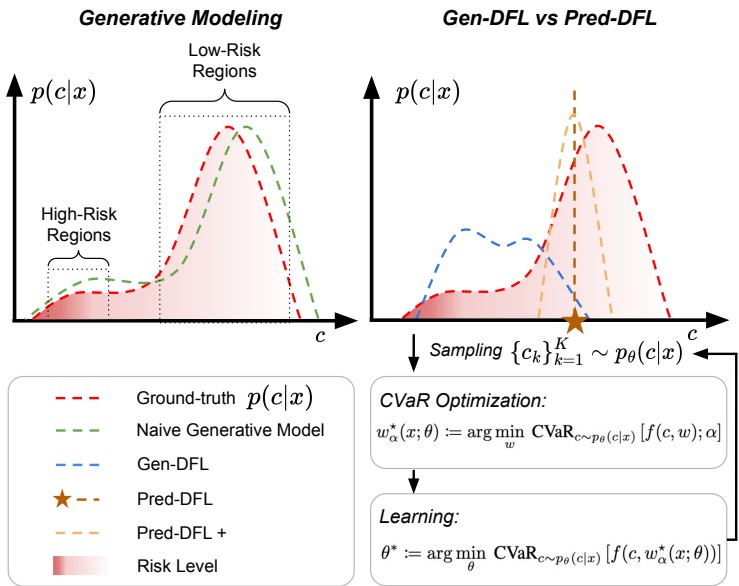

Figure 2: Unlike Pred-DFL, `Gen-DFL` leverages a generative model to capture $p(c|x)$ while incorporating the decision-making objective which emphasizes the high-risk region.

**Robust Optimization.** In some real-world applications, the expectation-based optimization in equation 2 may fail to provide reliable decisions under adverse conditions, potentially leading to severe consequences Ben-Tal et al. (2009); Beyer & Sendhoff (2007). To mitigate this risk, robust optimization (RO) Kouvelis & Yu (1997); Ben-Tal et al. (2009); Shalev-Shwartz & Wexler (2016) seeks decisions that perform well in the worst-case scenario within an uncertainty set $\mathcal{U}(x)$, by solving the min-max formulation below:

$$w^\star(x) := \arg\min_w \max_{c \in \mathcal{U}(x)} f(c, w). \tag{3}$$

This formulation ensures robustness against the most adverse realization of $c$, providing worst-case protection. However, it can be overly conservative, potentially leading to suboptimal decisions in typical scenarios. In many risk-sensitive applications, a more nuanced approach is required – one that balances robustness and flexibility by considering a broader range of adverse outcomes beyond just the extreme worst case Sarykalin et al. (2008). This has led to the development of alternative robust and risk-aware optimization frameworks, such as distributionally robust optimization (DRO) Gao et al. (2018); Zhu et al. (2022); Chen et al. (2025a) and conditional value-at-risk (CVaR) optimization Duffie & Pan (1997); Rockafellar et al. (2000); Rockafellar & Uryasev (2002), which offer a more refined trade-off between robustness and performance.

## 4 PROPOSED FRAMEWORK: GEN-DFL

This section presents the proposed decision-focused generative learning (`Gen-DFL`) framework. Specifically, we develop a novel decision-making paradigm, generate-then-optimize (GTO), designed for risk-sensitive decision problems. Our approach frames the problem as a conditional value-at-risk (CVaR) optimization, leveraging a generative model to produce plausible samples that capture the dynamics of high-risk regions. To effectively learn the generative model, we propose a new loss function that integrates both decision-focused learning and generative modeling objectives, ensuring that the generated samples not only reflect the underlying data distribution but also lead to robust, high-quality decisions. Figure 2 provides an overview of the proposed framework.

**Problem Setup.** We seek robust decisions that effectively manage risk by minimizing the percentiles of loss distributions. This approach has been widely adopted in risk-sensitive domains such as financial portfolio optimization, where regulatory frameworks often define risk management requirements

in terms of loss percentiles Sarykalin et al. (2008). A widely used measure for quantifying high-loss scenarios is conditional value-at-risk (CVaR) Duffie & Pan (1997); Rockafellar et al. (2000); Rockafellar & Uryasev (2002), which provides a characterization of tail risk by capturing the expected loss beyond a given percentile threshold. Formally, given a confidence level $\alpha$, CVaR is defined as:

$$\text{CVaR}[f(c, w); \alpha] = \mathbb{E}\left[f(c, w) \mid f(c, w) \geq \text{VaR}_\alpha\right], \tag{4}$$

where $\text{VaR}_\alpha$ represents the value-at-risk threshold; the probability of exceeding the threshold is at most $\alpha$. Our objective is to find the decision $w^\star$ that minimizes the costs in the worst-$\alpha\%$ of outcomes. This leads to the following risk-sensitive optimization formulation Krokhmal et al. (2002):

$$w^\star(x; \alpha) := \arg\min_w \text{CVaR}_{c \sim p(c|x)}[f(c, w); \alpha]. \tag{5}$$

We note that the $c$ is defined over the high-risk region of the distribution $p(c|x)$, allowing for a more flexible and probabilistic characterization of uncertainty compared to the "hard" uncertainty set used in equation 3. This formulation bridges robust and expectation-based optimization: ($i$) As $\alpha \to 0$, the problem reduces to robust optimization, focusing exclusively on the worst-case scenario in equation 3. ($ii$) As $\alpha \to 1$, it converges to standard expectation-based optimization in equation 1, minimizing the expected cost across all possible outcomes. Thus, our approach generalizes robust optimization by ensuring resilience against adverse outcomes beyond a single worst-case scenario, balancing conservatism and probabilistic risk awareness in decision-making.

### 4.1 GENERATE-THEN-OPTIMIZE

To solve equation 5, we introduce a novel generate-then-optimize (GTO) paradigm, which leverages generative modeling to approximate the risk-sensitive optimization problem. Conventional decision-focused learning (Pred-DFL) relies on a point estimate $\hat{c}$ of the optimization parameters. While effective in some cases, this approach fails to capture the full distribution $p(c|x)$, particularly in high-dimensional settings, making it inadequate for risk-sensitive applications where adverse outcomes must be explicitly considered. Moreover, point estimates are only appropriate when the objective function is linear, as the optimization problem in such cases depends solely on the expected value of $c$, making variance and higher-order moments irrelevant.

To overcome these limitations, we replace deterministic predictions with a generative model, capturing the full risk distribution. This allows us to account for uncertainty in a data-driven manner, ensuring that risk-sensitive scenarios are explicitly considered. The optimization problem is then solved using sample-average approximation (SAA) Pagnoncelli et al. (2009); Kim et al. (2015); Emelogu et al. (2016). Formally, we aim to optimize:

$$w_\theta^\star(x; \alpha) := \arg\min_w \text{CVaR}_{c \sim p_\theta(c|x)}[f(c, w); \alpha]. \tag{6}$$

Unlike traditional RO, which requires a pre-defined uncertainty set $\mathcal{U}(x)$ – often leading to overly conservative or restrictive formulations – our approach treats uncertainty as a learnable distribution. Specifically, we model $p_\theta(c|x)$ using a generative model parameterized by $\theta$, allowing it to adaptively capture risk-sensitive regions based on empirical data. This approach provides a more nuanced and adaptive approach to uncertainty modeling, ensuring that decisions are informed by the full distribution of possible outcomes rather than rigid, pre-specified constraints.

We emphasize that the proposed `Gen-DFL` framework is model-agnostic and does not rely on a specific generative modeling choice. In this work, we adopt conditional normalizing flows (CNFs) Winkler et al. (2019) to model the conditional distribution $p(c|x)$ due to their flexibility and tractable likelihood evaluation. CNFs transform a simple base distribution $p_Z(z)$ (*e.g.*, Gaussian) into a complex target distribution via an invertible mapping $g_\theta : \mathcal{C} \to \mathcal{Z}$, where $\mathcal{C}, \mathcal{Z}$ are the supports of the resulting distribution and the base distribution. This enables the representation of arbitrarily complex, high-dimensional distributions. This transformation follows the change-of-variables formula Tabak & Turner (2013); Papamakarios et al. (2021): $p_\theta(c|x) = p_Z(g_\theta(c; x)) \left| \frac{\det \partial g_\theta(c; x)}{\partial c} \right|$. This expressiveness enables our model to generate samples that capture both typical and high-risk scenarios, improving robustness in decision-making under CVaR.

### 4.2 DECISION-FOCUSED GENERATIVE LEARNING

We now present the `Gen-DFL` framework, which provides a decision-focused solution to the GTO problems in a unified end-to-end learning pipeline. For simplicity, we denote the optimal decision

obtained from our model $w_\theta^\star(x; \alpha)$ in equation 6 as $w_\theta^\star$, omitting $x$ and $\alpha$. Similar to other DFL frameworks, `Gen-DFL` consists of two alternating steps:

1. *Generate-Then-Optimize*: Generate samples $\{c_k\}_{k=1}^K$ using conditional generative model (CGM) $p_\theta(c|x)$ and solve equation 6 for the optimal decision via SAA.
2. *Model Learning*: Given the resulting decision $w_\theta^\star$, update the generative model parameters by jointly minimizing the generative loss and the decision cost under $w_\theta^\star$.

A detailed description of the learning procedure is provided in Algorithm 1 in Appendix C.1. Below, we elaborate on key components of our framework.

**Regret in CVaR.** Unlike Pred-DFL, where the decision cost is computed as the regret for a single pair $(\hat{c}, c)$, in our stochastic optimization problem, the parameter $c$ follows a distribution, requiring regret to be evaluated over all possible realizations of $c$. Moreover, in robust decision-making, we seek to minimize decision costs based on the worst-$\alpha\%$ outcomes, rather than the full distribution. To capture this, we define regret using CVaR:

$$\text{Regret}_{\theta,p}(x; \alpha) := \text{CVaR}_{p(c|x)}\Big[f(c, w_\theta^\star) - f(c, w^\star); \alpha\Big],$$

where $w^\star := \arg\min_w \text{CVaR}_{c\sim p(c|x)}[f(c, w); \alpha]$ is the optimal decision under the true distribution. The parameter $\alpha$ controls the level of risk sensitivity: The lower values of $\alpha$ emphasize the worst-case outcomes, making decisions more conservative. When $\alpha = 1$, it recovers the standard expected regret across all realizations: $\mathbb{E}_{c\sim p(c|x)}[f(c, w_\theta^\star) - f(c, w^\star)]$.

**Gen-DFL Loss.** In practice, the true data distribution $p(c|x)$ is typically inaccessible, making direct regret evaluation infeasible. To address this challenge, we introduce an auxiliary model $q(c|x)$, trained on available data to approximate $p(c|x)$. Once learned, $q(c|x)$ remains fixed and serves as a proxy distribution to compute the estimated $\text{Regret}_{\theta,q}(x, \alpha)$ and the corresponding surrogate loss function $\ell(\theta; \alpha, q)$. This enables regret evaluation even when the true distribution is not directly observable. The training objective for `Gen-DFL` can then be formulated as the aggregated regret across all inputs $x$ with an additional regularization term to ensure stability in generative modeling:

$$\ell_{\texttt{Gen-DFL}}(\theta; q, \alpha) := \beta \cdot \mathbb{E}_x[\text{Regret}_{\theta,q}(x; \alpha)] + \gamma \cdot \ell_{\text{gen}}(\theta), \tag{7}$$

where $\ell_{\text{gen}}(\theta)$ is the generative model loss (*e.g.*, negative log-likelihood, evidence lower bound (ELBO) for variational autoencoders Kingma (2013), or score-matching loss for diffusion models Ho et al. (2020)). Here, $\beta$ and $\gamma$ are hyperparameters that balance the decision-focused regret loss and the generative model loss. The generative loss term $\ell_{\text{gen}}(\theta)$ acts as a regularization, preventing the learned generative model from deviating excessively from the true data distribution, ensuring reliable sample generation for decision-making.

**Surrogate Loss Function.** When training DFL models with respect to the decision loss, it is necessary to backpropagate errors through the decision variable. However, this requires computing the partial derivatives $\frac{\partial w_\theta^\star}{\partial c}$, which often involves complex dependency chains. Inspired by Mulamba et al. (2020), we propose a surrogate contrastive loss in our Gen-DFL setting to address the challenge of differentiating through the combinatorial optimization mapping:

$$\ell_{\texttt{Gen-DFL}}(\theta; \alpha) = \beta \cdot \mathbb{E}_x\Big[ \sum_{w^s \in S} \Big(\text{CVaR}_{p_\theta(c|x)}\big[f(c, w^s) - f(c, w^\star); \alpha\big]\Big)\Big] + \gamma \cdot \ell_{gen}(\theta),$$

where $w^\star$ is the target solution, and the negative samples $w^s \in S \subset \mathcal{W} \setminus \{w^\star\}$ are a subset of solutions that differ from the target solution in practice.

## 5 THEORETICAL ANALYSIS

This section provides an analysis of the validity of our sample-based regret estimation method and compares `Gen-DFL` and traditional Pred-DFL across different problem settings by examining their regret bounds. Our analysis reveals that as the complexity of the optimization problem increases, whether due to higher dimensionality, greater variance in the data, or a more nonlinear objective

function, `Gen-DFL`'s advantage over Pred-DFL becomes more pronounced, leading to improved decision quality in the most challenging practical settings.

We first derive the bound for the loss difference $|\ell(\theta; p, \alpha) - \ell(\theta; q, \alpha)|$, comparing the loss function $\ell(\theta; p, \alpha)$ under the ground-truth distribution $p(c|x)$ with the surrogate loss $\ell(\theta; q, \alpha)$ computed using the proxy model $q(c|x)$. The proofs of the theorems can be found in Appendix A.1.

**Theorem 5.1.** *Under the assumption that the objective function $f(c, w)$ is $L_f$-Lipschitz continuous with respect to $c$ for a fixed decision variable $w$, the gap between $\ell(\theta; p, \alpha)$ and $\ell(\theta; q, \alpha)$ is bounded by*

$$|\ell(\theta; p, \alpha) - \ell(\theta; q, \alpha)| \leq K_q \cdot \mathbb{E}_x\left[\mathcal{W}(p(c|x), q(c|x))\right],$$

*where $\mathcal{W}(p(c|x), q(c|x))$ is the Wasserstein-1 distance between $p(c|x)$ and $q(c|x)$ and $K_q$ is some constant.*

The theorem above implies that the surrogate loss provides a valid approximation to the original loss function, provided the proxy model $q(c|x)$ can estimate the ground-truth $p(c|x)$ well. The bound is directly proportional to the $\mathcal{W}(p(c|x), q(c|x))$, which quantifies the discrepancy between these distributions. We now establish the conditions under which `Gen-DFL` outperforms Pred-DFL. To facilitate our analysis, we first introduce the following two definitions.

**Definition 5.2.** Let $p(c|x)$ denote the true conditional distribution of $c$, and let $p_\theta(c|x)$ be the generative model. We define $Q_c$ to be the "worst $\alpha\%$ tail" representative for $c$ under $p(c|x)$ based on the target decision $w^\star$. Formally,

$$Q_c[\alpha] := \mathbb{E}[c \mid f(c, w^\star) \geq \text{VaR}_\alpha].$$

**Definition 5.3.** Given the target decision $w^\star$ and the decisions found by Pred-DFL ($w^\star_{\text{pred}}$) and `Gen-DFL` ($w^\star_\theta$), we can define the regret of Pred-DFL and `Gen-DFL` as:

$$R_{\text{pred}}(x; \alpha) = f(Q_c[\alpha], w^\star_{\text{pred}}) - f(Q_c[\alpha], w^\star), \ R_{\text{gen}}(x; \alpha) = \text{CVaR}_{p(c|x)}\left[f(c, w^\star_\theta) - f(c, w^\star); \alpha\right].$$

Next, we develop a regret bound that quantifies the performance gap between `Gen-DFL` and Pred-DFL, incorporating data variance and optimization complexity, such as the dimensionality of the parameter space and the risk-sensitive level. The proof can be found in Appendix A.3.

**Theorem 5.4.** *Let $g : \mathcal{X} \to \mathcal{C}$ be the predictor in Pred-DFL. Assume the objective function $f(c, w)$ is Lipschitz continuous for any $c, w$. There exists some constants $L_w, L_c, \kappa_1, \kappa_2, \kappa_3$ such that the following upper-bound holds for the aggregated regret gap $\mathbb{E}_x|\Delta R(x)|$:*

$$\mathbb{E}_x[\Delta R(x)] \leq \mathbb{E}_x\left[\frac{2L_w}{\alpha}\left[\kappa_1 \mathcal{W}(p_\theta, p) + \kappa_2 \|\text{Bias}[g]\|\right] + \left(\frac{2L_w}{\alpha}\kappa_3 + 2L_c\right)\sqrt{\|\text{Var}[c \mid x]\|}\right.$$
$$\left. + \text{CVaR}_{p(c|x)}[\|\text{Bias}[g(x)]\|; \alpha]\right], \text{ where } \Delta R(x) = R_{\text{pred}}(x; \alpha) - R_{\text{gen}}(x; \alpha).$$

The above results reveal how the following three factors affect the performance gap between `Gen-DFL` and Pred-DFL: (*i*) Variance of the parameter space $\|\text{Var}[c|x]\|$: Higher variance in $c$ conditioned on $x$ increases uncertainty and amplifies the difficulty of accurately approximating the objective. Pred-DFL, which relies on point estimates from $g(x)$, struggles in high-variance settings. In contrast, `Gen-DFL` benefits from modeling the full distribution $p(c|x)$, capturing the variability and structure needed for robust decision-making under uncertainty; (*ii*) Dimensionality of the parameter space, including $d_c$ and $d_x$: As the dimensionality increases, the estimation error of the predictor in Pred-DFL grows at a rate of $\mathcal{O}(\sqrt{(d_x + d_c)/n}/\alpha)$ (see proof in Appendix A.5), making it increasingly difficult to obtain reliable point estimates; (*iii*) Risk level $\alpha$: The inverse dependence of the estimation error on $\alpha$ implies that smaller values of $\alpha$ make quantile regression more challenging for Pred-DFL, as data in the tail regions of the worst $\alpha\%$ outcomes become increasingly sparse. This leads to a larger bias in $g(x)$ for smaller $\alpha$. In contrast, `Gen-DFL` leverages a generative model to capture the full conditional distribution $p(c|x)$. Together, these insights demonstrate that `Gen-DFL` offers significant advantages over Pred-DFL in complex, high-dimensional, and risk-sensitive scenarios.

| Task | | Pairwise | Listwise | NCE | MAP | SPO+ | Diff-DRO | 2Stage (PTO) | Gen-DFL |
|------|---|----------|----------|-----|-----|------|----------|--------------|---------|
| **Portfolio** | Deg-2 | 11.48±(0.50) | 22.87±(1.11) | 8.57±(0.48) | 8.88±(0.34) | 6.92±(0.26) | 8.30±(0.36) | 16.90±(0.55) | **3.71±(0.18)** |
| | Deg-4 | 11.16±(0.32) | 20.70±(1.19) | 7.81±(0.52) | 8.43±(0.65) | 7.23±(0.60) | 7.41±(0.67) | 14.89±(0.63) | **3.81±(0.22)** |
| | Deg-6 | 11.54±(0.78) | 18.57±(0.87) | 8.69±(0.61) | 8.51±(0.38) | 7.01±(0.26) | 8.56±(0.71) | 16.02±(0.78) | **4.31±(0.32)** |
| | Deg-8 | 10.44±(0.36) | 21.92±(0.95) | 7.93±(0.40) | 8.90±(0.48) | 6.98±(0.98) | 8.65±(0.52) | 16.17±(0.60) | **3.59±(0.31)** |
| **Knapsack** | Deg-2 | 34.93±(9.37) | 27.03±(8.43) | 24.75±(7.87) | 35.54±(4.70) | 21.90±(7.46) | 19.63±(4.5) | 20.27±(9.46) | **17.60±(3.38)** |
| | Deg-4 | 38.32±(4.44) | 26.37±(3.03) | 23.43±(4.94) | 46.87±(14.43) | 20.37±(5.18) | 18.45±(3.81) | 16.58±(3.68) | **15.21±(3.75)** |
| | Deg-6 | 33.85±(8.24) | 24.50±(1.19) | 20.07±(10.76) | 40.33±(5.63) | **17.45±(7.2)** | 17.51±(5.20) | 21.66±(6.46) | 17.91±(2.44) |
| | Deg-8 | 33.25±(6.48) | 20.38±(6.70) | 22.36±(7.89) | 34.07±(6.66) | 22.90±(11.48) | 21.48±(6.28) | 21.13±(7.40) | **19.29±(3.75)** |
| **Shortest Path** | Deg-2 | 8.30±(2.35) | 2.65±(0.25) | 9.59±(0.75) | 12.92±(3.63) | 3.23±(0.72) | 2.91±(0.93) | 10.07±(1.2) | **1.87±(0.20)** |
| | Deg-4 | 18.91±(5.30) | 12.19±(1.04) | 42.87±(2.57) | 52.47±(6.49) | 28.73±(11.23) | 11.78±(2.89) | 22.44±(2.84) | **3.64±(0.43)** |
| | Deg-6 | 29.63±(7.20) | 33.15±(4.60) | 68.94±(6.79) | 94.46±(10.91) | 26.46±(9.31) | 23.76±(4.21) | 38.64±(2.3) | **6.52±(0.71)** |
| | Deg-8 | 63.61±(18.82) | 51.65±(13.77) | 139.09±(22.08) | 173.17±(36.28) | 81.78±(21.82) | 39.81±(5.46) | 45.75±(5.10) | **13.36±(2.59)** |
| **Energy** | | 1.65±(0.23) | 1.67±(0.17) | 1.69±(0.13) | 1.59±(0.11) | 1.56±(0.11) | 1.49±(0.12) | 1.91±(0.22) | **1.09±(0.09)** |
| **COVID Resource** | | 17.91±(1.85) | 16.83±(1.07) | 16.48±(2.25) | 16.59±(3.04) | 17.94±(3.29) | **16.41±(3.8)** | 18.46±(3.2) | 16.86±(4.62) |

Table 1: Comparison of decision quality (average relative regret, $\downarrow$, lower is better) across tasks in high-variance settings ($\sigma = 20$).

# 6 EXPERIMENTS

## 6.1 EXPERIMENTAL SETUP

We evaluate the proposed framework on three synthetic optimization problems including Portfolio Management, Fractional Knapsack, and Shortest-Path. We also consider two real-world tasks, namely an Energy Management Problem dataset Ifrim et al. (2012); Simonis et al. (1999) and a COVID-19 resource allocation problem adopted from Mandi et al. (2022); Kong et al. (2022).

**Synthetic Data.** We evaluate our approaches on synthetic benchmarks (Portfolio, Knapsack, and Shortest-Path), adopting the data-generation process from Elmachtoub & Grigas (2022). We first overview the optimization setup for the Portfolio problem. In the Portfolio problem, optimization parameters $c$ represents the asset prices and the dimension of $c_i$ is the number of assets. Our non-linear, risk-sensitive Portfolio problem is then formulated as:

$$w^{\star}(x; \alpha) := \arg\min_{w} \mathrm{CVaR}_{p(c|x)}[-c^T w + w^T \Sigma w; \alpha], \text{ s.t. } w \in [0, 1]^n, \mathbf{1}^T w \leq 1, \tag{8}$$

where $\Sigma = LL^T + (0.01\sigma)^2 I$ is the covariance among the asset prices $c$, and the quadratic term $w^T \Sigma w$ reflects the amount of risk. The configurations of our synthetic experiments include the training size, feature dimension $d_x$, polynomial degree, and the noise scale $\sigma$ that reflects the amount of variance in the parameter space and the non-linearity of the above stochastic optimization, since, by our construction, $\sigma$ would affect the magnitude of the quadratic term $w^T \Sigma w$. The problem setup and model configurations for the Fractional Knapsack and Shortest-Path problem are similar to that of Portfolio. Full details of the data synthesis process and the corresponding optimization formulation for each experiment are provided in Appendix B.

**Real Data.** For the real data experiments, we consider a real-world Energy-cost Aware Scheduling problem and a COVID-19 resource allocation problem that we adopt from Mandi et al. (2022); Kong et al. (2022). In the Energy-cost Aware Scheduling problem, we consider a demand response program in which an operator schedules electricity consumption $p_t \in \mathbb{R}^{24}$ over a time horizon $t \in \Omega_t$. The objective is to minimize the total cost of electricity while adhering to operational constraints. In the COVID-19 resource allocation problem, we focus on the problem of allocating the number of hospital beds $w \in \mathbb{R}^7$ for the next seven days based on the forecasted number of hospitalized patients $c \in \mathbb{R}^7$. The details of optimization problem in each experiment can be found in Appendix B.

**Model Configuration.** The hyperparameters in our learning algorithm include the decision cost weight $\beta$ and the negative log-likelihood weight $\gamma$ in equation 7, which serves as regularization. We introduce an additional hyperparameter $\beta$ in our experiment to study how different magnitude of DFL loss will affect the model's performance. We set $\gamma = 1$ across all experiments and study the effect of different $\beta$ values on Gen-DFL's performance (Figure 3). When $\beta = 0$, the loss reduces to that of a standard generative model, only fitting data without considering decision costs, which results in the worst regret in all risk-sensitive settings. Increasing $\beta$ improves downstream decision quality across all risk levels. Full hyperparameter details are provided in Appendix C.2.

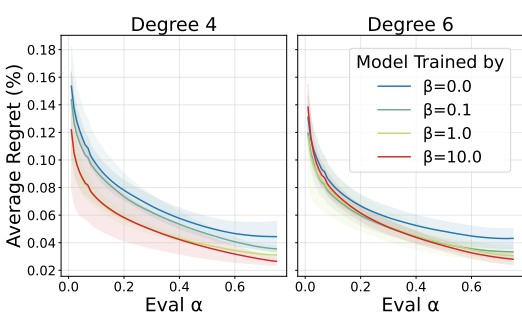 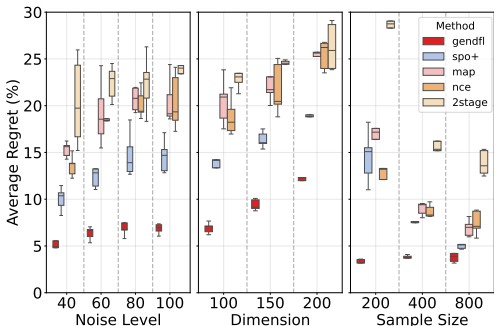

Figure 3: Decision quality against different risk-sensitive regions vs various $\beta$ in Portfolio problem.

Figure 4: Decision quality in the portfolio problem under different settings ($\downarrow$ lower is better).

**Baseline Methods.** We evaluate the performance of `Gen-DFL` against various state-of-the-art Pred-DFL baselines across all tasks. Specifically, we compare against Smart-Predict-Then-Optimize (SPO+) Elmachtoub & Grigas (2022), contrastive loss-based Pred-DFL models (NCE, MAP) Mulamba et al. (2020), ranking-based Pred-DFL models Mandi et al. (2022), and the recently proposed Pred-DFL approach with differentiable Distributionally Robust Optimization layers, which we refer to as Diff-DRO Ma et al. (2024). These baselines represent a range of decision-focused learning strategies, differing in their loss formulations and optimization objectives. The main results of our comparison are summarized in Table 1.

We evaluate the decision quality of different models on various tasks in terms of the average relative regret, $\mathbb{E}_x \left[ \frac{\text{CVaR}_{p(c|x)}[f(c,\hat{w}^\star) - f(c,w^\star);\alpha]}{\mathbb{E}_{p(c|x)}[f(c,w^\star)]} \right] \times 100\%$. where lower $\alpha$ indicates greater risk sensitivity. For our real data experiment, we will first train a proxy model $q(c|x)$ given the data, which will then be used to evaluate the average relative regret during evaluation. We set $\alpha = 1$ when we compare against the baseline models since the above metric is equivalent to the standard relative regret used in previous Pred-DFL literature, which makes the comparison fair.

## 6.2 RESULTS

**Comparison with Baselines.** Table 1 presents the comparative performance of `Gen-DFL`, Pred-DFL, and the two-stage method across different problem settings. `Gen-DFL` consistently outperforms baseline methods, reducing regret by up to 58.5% compared to Diff-DRO and up to 48.5% compared to SPO+ in Portfolio tasks. `Gen-DFL`'s advantage is particularly pronounced in high-dimensional tasks like Shortest-Path (Deg-8), where it achieves a remarkable 83.7% reduction in regret over SPO+ (13.36 vs. 81.78). This demonstrates `Gen-DFL`'s ability to overcome the curse of dimensionality by effectively capturing the distributional structure of $p(c|x)$ rather than relying on point estimates.

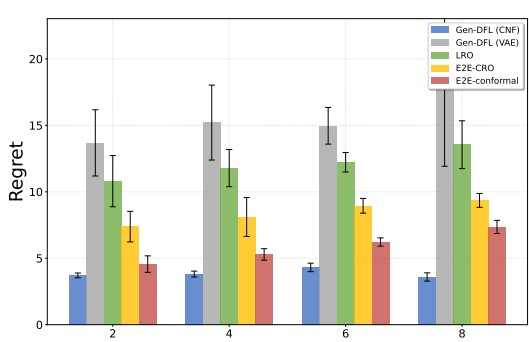

Figure 5: Comparison with conventional RO under varying polynomial degree in Portfolio problem.

Conversely, in Knapsack (Deg-2), `Gen-DFL`'s improvements over SPO+ and Diff-DRO are more moderate (19.6% and 10.3% respectively), suggesting that the benefits of generative modeling are especially significant in problems where uncertainty is highly non-linear or where high-dimensional interactions dominate the optimization landscape. The statistical significance tests of Table 1 can be found in the Appendix E.

Figure 4 illustrates the impact of variance $\text{Var}[c|x]$, problem dimensionality, and training size on model performance. `Gen-DFL` demonstrates robustness across all variance levels ($\sigma \in [40, 60, 80, 100]$), effectively capturing the full conditional distribution $p(c|x)$, unlike Pred-DFL models, which rely on less expressive predictors and are more sensitive to variance. As dimensionality increases, baseline methods suffer from the curse of dimensionality, leading to higher regret. In contrast, `Gen-DFL`

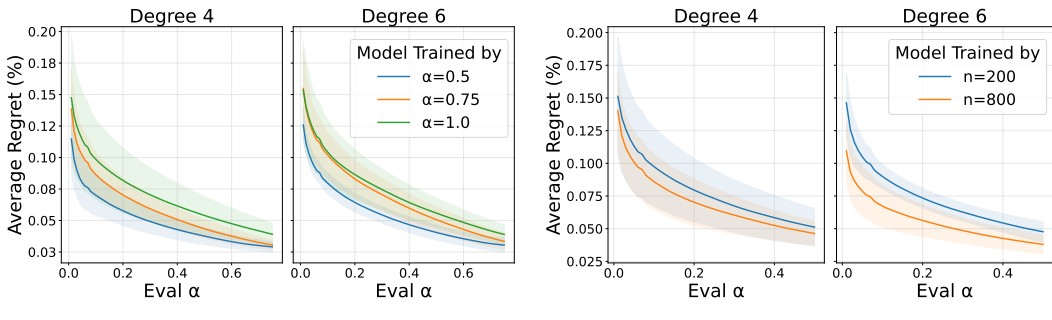

(a) Training $\alpha$ vs. risk level $\alpha$.     (b) Number of generated samples vs. risk level $\alpha$.

Figure 6: Portfolio problem ablation results.

maintains superior performance by learning the structural complexity of $p(c|x)$, as predicted in Theorem 5.4. Additionally, while Pred-DFL performance deteriorates with smaller training sizes due to increased predictor bias, `Gen-DFL` remains stable by effectively modeling the underlying distribution. The quadratic term in the objective further amplifies the non-linearity in high-variance settings, demonstrating `Gen-DFL`'s adaptability to complex optimization problems.

We further compare our `Gen-DFL` with conventional data-driven Robust Optimization methods (*e.g.*, LRO Wang et al. (2025), E2E-CRO Chenreddy & Delage (2024), E2E-Conformal Yeh et al. (2024)) in Figure 5. Instead of learning fixed-geometry uncertainty sets for a hard min-max problem, `Gen-DFL` learns a full generative model to directly capture complex uncertainties and minimizes the Conditional Value-at-Risk (CVaR). We also did an ablation study on the choice of the generative models which shows that the power of our method lies in this core paradigm; while a Conditional Normalizing Flow (CNF) architecture outperforms a Variational Autoencoder (VAE) (of which we attribute to the advantages of exact likelihood training over the ELBO approximation), the primary contribution remains the development and validation of the `Gen-DFL` framework itself. We thus leave the integration of other generative models or optimization schemes to future work.

**Risk-sensitive Settings.**   We also evaluate `Gen-DFL` under various risk-sensitive settings (indicated by the "Eval $\alpha$" on the x-axis, where smaller "Eval $\alpha$" indicates that we are evaluating under the higher-risk regions) using CVaR, which measures the decision quality (in terms of regret) over the worst-$\alpha\%$ of outcomes. Figure 6a shows that models trained with smaller $\alpha$ (*e.g.*, $\alpha = 0.5$) outperform those trained with larger $\alpha$ (*e.g.*, $\alpha = 1.0$), demonstrating better adaptation to adverse outcomes. The performance gap widens as risk sensitivity increases, confirming that smaller $\alpha$ enhances robustness while larger $\alpha$ prioritizes average-case performance. To further assess stability, we examine the impact of sample size in the sample-average-approximation step (Figure 6b). Increasing generated samples consistently improves decision quality across all risk levels, reinforcing the importance of uncertainty modeling in `Gen-DFL`. These results highlight `Gen-DFL`'s flexibility, making it particularly effective in high-stakes, risk-sensitive environments.

## 7   CONCLUSION

We presented `Gen-DFL`, a novel decision-focused learning framework that leverages generative modeling to solve robust decision-making problems under various risk-sensitive settings and provided a thorough theoretical analysis demonstrating the performance gain under various high-risk decision-making problems, verified by comprehensive experiments.

## ETHICS STATEMENT

This paper presents work whose goal is to advance the field of Machine Learning. There are many potential societal consequences of our work, none of which we feel must be specifically highlighted here.

## REPRODUCIBILITY STATEMENT

We include our full implementation at `https://github.com/kingofspace0wzz/gen_dfl`.

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

# A  THEOREMS AND PROOFS

## A.1  SURROGATE LOSS FUNCTION

In this subsection, we present a theoretical bound on the gap between the loss function $\ell(\theta; p, \alpha)$ under the ground-truth distribution $p(c \mid x)$ and the surrogate loss $\ell(\theta; q, \alpha)$ under a proxy distribution $q(c \mid x)$ that approximates $p(c \mid x)$.

**Theorem A.1.** *Let $p(c|x)$ be the ground-truth distribution and $q(c|x)$ be a surrogate distribution that approximates $p(c|x)$.*

*Under the assumption that the objective function $f(c, w)$ is $L_f$-Lipschitz continuous with respect to $c$ for a fixed decision variable $w$, the gap between the loss function $\ell(\theta; p, \alpha)$ and the surrogate loss $\ell(\theta; q, \alpha)$ is bounded by*

$$|\ell(\theta; p, \alpha) - \ell(\theta; q, \alpha)| \leq K_q \cdot \mathbb{E}_x \left[ \mathcal{W}(p(c|x), q(c|x)) \right],$$

*where $\mathcal{W}(p(c|x), q(c|x))$ is the Wasserstein distance between $p(c|x)$ and $q(c|x)$ and $K_q$ is some constant.*

*Proof.* First, by linearity of expectation and the triangle inequality, we see that

$$|\ell(\theta; p, \alpha) - \ell(\theta; q, \alpha)| \leq \mathbb{E}_x |A - B|, \tag{9}$$

where

$$A = |\text{CVaR}_{p(c|x)}[f(c, w_\theta^\star)] - \text{CVaR}_{p(c|x)}[f(c, w^\star)]|, \qquad B = |\text{CVaR}_{q(c|x)}[f(c, w_\theta^\star)] - \text{CVaR}_{q(c|x)}[f(c, w^\star)]|.$$

For simplicity, we omit $\alpha$ inside CVaR for now.

Then, we can begin by examining the gap between the expectations under $p(c|x)$ and $q(c|x)$ for a fixed context $x$.

By the reverse triangle inequality ($||x| - |y|| \leq |x - y|$), we have

$$|A - B| \leq \left| \text{CVaR}_{p(c|x)}[f(c, w^\star)] - \text{CVaR}_{q(c|x)}[f(c, w^\star)] \right| + \left| \text{CVaR}_{p(c|x)}[f(c, w_\theta^\star)] - \text{CVaR}_{q(c|x)}[f(c, w_\theta^\star)] \right|.$$

Let's define $g(c) = f(c, w^\star)$ and $h(c) = f(c, w_\theta^\star)$. By assumption, $f(c, w)$ is $L_f$-Lipschitz continuous with respect to $c$, which implies that that $g(c)$ and $h(c)$ are also $L_f$-Lipschitz. Hence, by the Kantorvorich-Rubinstein duality for the Wasserstein distance, we have,

$$\mathcal{W}(p(c|x), q(c|x)) = \sup_{\|g\|_{\text{Lip}} \leq 1} \left| \mathbb{E}_{c \sim p(c|x)}[g(c)] - \mathbb{E}_{c \sim q(c|x)}[g(c)] \right| = \sup_{\|h\|_{\text{Lip}} \leq 1} \left| \mathbb{E}_{c \sim p(c|x)}[h(c)] - \mathbb{E}_{c \sim q(c|x)}[h(c)] \right|,$$

where the supremum is over all functions $g, h$ that are 1-Lipschitz.

By definition of CVaR, we can see that,

$$\left| \text{CVaR}_{p(c|x)}[f(c, w^\star)] - \text{CVaR}_{q(c|x)}[f(c, w^\star)] \right| \leq \sup_{\|h\|_{\text{Lip}} \leq 1} \left| \mathbb{E}_{c \sim p(c|x)}[h(c)] - \mathbb{E}_{c \sim q(c|x)}[h(c)] \right|.$$

Again, using the assumption that $g(c)$ and $h(c)$ are also $L_f$-Lipschitz, we can bound the gap in (2) by

$$\left| \text{CVaR}_{p(c|x)}[f(c, w^\star)] - \text{CVaR}_{q(c|x)}[f(c, w^\star)] \right| + \left| \text{CVaR}_{p(c|x)}[f(c, w_\theta^\star)] - \text{CVaR}_{q(c|x)}[f(c, w_\theta^\star)] \right| \leq 2 L_f \mathcal{W}(p(c|x), q(c|x)).$$

Finally, taking the expectation over $x$ on both sides and using equation (1) and set the constant $K_q = 2 L_f$, we get:

$$|\ell(\theta; p, \alpha) - \ell(\theta; q, \alpha)| \leq K_q \cdot \mathbb{E}_x \left[ \mathcal{W}(p(c|x), q(c|x)) \right].$$

This completes the proof. □

## A.2 CVAR/QUANTILE REGRESSION

**Theorem A.2** (Finite-Sample Bound for CVaR Estimation). *Suppose $Y$ takes values in the interval $[m, M]$. Let $\widehat{\text{CVaR}}_\alpha$ be the empirical estimator derived from*

$$\hat{\phi}_n(\eta) \;=\; \eta \;+\; \frac{1}{\alpha}\frac{1}{n}\sum_{i=1}^n (Y_i - \eta)_+, \quad \widehat{\text{CVaR}}_\alpha \;=\; \inf_{\eta \in \mathbb{R}} \hat{\phi}_n(\eta),$$

*where $(y - \eta)_+ := \max\{y - \eta, 0\}$ and $Y_1, \ldots, Y_n$ are i.i.d. samples of $Y$. Then there is a universal constant $C > 0$ such that for all $\delta > 0$, with probability at least $1 - \delta$,*

$$\left| \widehat{\text{CVaR}}_\alpha \;-\; \text{CVaR}_\alpha(Y) \right| \;\leq\; C\,\frac{(M - m)}{\alpha}\,\sqrt{\frac{\ln(1/\delta)}{n}}.$$

*In other words, the estimation error for $\text{CVaR}_\alpha$ converges on the order of $\sqrt{\ln(1/\delta)/n}$ as $n$ grows.*

**Remark A.3.** Here, $\text{CVaR}_\alpha(Y) = \mathbb{E}[\, Y \mid Y \leq \text{VaR}_\alpha(Y)\,]$, and

$$\text{VaR}_\alpha(Y) \;=\; \inf\{\, t : \Pr(Y \leq t) \;\geq\; \alpha\}.$$

The key step in the proof is the Rockafellar–Uryasev identity,

$$\text{CVaR}_\alpha(Y) \;=\; \inf_{\eta \in \mathbb{R}}\Big\{\eta \;+\; \tfrac{1}{\alpha}\,\mathbb{E}\big[(Y - \eta)_+\big]\Big\},$$

combined with uniform convergence arguments (*e.g.*, Hoeffding or Rademacher complexity bounds).

*Proof.* **Step 1: Rockafellar–Uryasev Representation.**

Recall the identity (Rockafellar–Uryasev):

$$\text{CVaR}_\alpha(Y) \;=\; \min_{\eta \in \mathbb{R}}\Big(\eta \;+\; \frac{1}{\alpha}\,\mathbb{E}\big[(Y - \eta)_+\big]\Big).$$

Set

$$\phi(\eta) \;=\; \eta \;+\; \frac{1}{\alpha}\,\mathbb{E}[(Y - \eta)_+].$$

Then $\text{CVaR}_\alpha(Y) = \min_{\eta \in \mathbb{R}} \phi(\eta)$.

**Step 2: Empirical Estimator.**

Given i.i.d. samples $Y_1, \ldots, Y_n$, define the empirical counterpart

$$\hat{\phi}_n(\eta) \;=\; \eta \;+\; \frac{1}{\alpha}\frac{1}{n}\sum_{i=1}^n (Y_i - \eta)_+,$$

and let

$$\widehat{\text{CVaR}}_\alpha \;=\; \min_{\eta \in \mathbb{R}} \hat{\phi}_n(\eta).$$

Similarly, let $\eta^* \in \arg\min_\eta \phi(\eta)$ and $\hat{\eta}_n \in \arg\min_\eta \hat{\phi}_n(\eta)$.

**Step 3: Uniform Convergence.**

Observe that

$$|\hat{\phi}_n(\eta) - \phi(\eta)| \;=\; \left|\frac{1}{\alpha}\Big(\tfrac{1}{n}\sum_{i=1}^n (Y_i - \eta)_+ - \mathbb{E}[(Y - \eta)_+]\Big)\right| \;\leq\; \frac{1}{\alpha}\sup_{\eta \in \mathbb{R}}\left|\tfrac{1}{n}\sum_{i=1}^n f_\eta(Y_i) \;-\; \mathbb{E}[f_\eta(Y)]\right|,$$

where $f_\eta(y) := (y - \eta)_+$ is bounded by $(M - m)$ if $y \in [m, M]$. By standard Hoeffding (or VC / Rademacher) arguments, with probability $\geq 1 - \delta$,

$$\sup_{\eta \in \mathbb{R}}\left|\tfrac{1}{n}\sum_{i=1}^n (Y_i - \eta)_+ \;-\; \mathbb{E}[(Y - \eta)_+]\right| \;\leq\; C_1\,(M - m)\,\sqrt{\frac{\ln(1/\delta)}{n}}$$

for some universal constant $C_1 > 0$. Hence,

$$\sup_{\eta \in \mathbb{R}} \left| \hat{\phi}_n(\eta) - \phi(\eta) \right| \ \leq \ \frac{C_1 \, (M - m)}{\alpha} \sqrt{\frac{\ln(1/\delta)}{n}} \ =: \varepsilon_n.$$

**Step 4: Error Between Minimizers.**

By definition of $\hat{\eta}_n$ and $\eta^*$,

$$\hat{\phi}_n(\hat{\eta}_n) \ \leq \ \hat{\phi}_n(\eta^*).$$

Also,

$$\phi(\hat{\eta}_n) - \phi(\eta^*) \ \leq \ \left[ \hat{\phi}_n(\hat{\eta}_n) - \phi(\hat{\eta}_n) \right] \ + \ \left[ \hat{\phi}_n(\eta^*) - \phi(\eta^*) \right] \ \leq \ 2 \, \varepsilon_n.$$

Thus

$$\phi(\hat{\eta}_n) \ \leq \ \phi(\eta^*) + 2 \, \varepsilon_n \implies \hat{\phi}_n(\hat{\eta}_n) \ = \ \phi(\hat{\eta}_n) + \left[ \hat{\phi}_n(\hat{\eta}_n) - \phi(\hat{\eta}_n) \right] \ \leq \ \phi(\eta^*) + 3 \, \varepsilon_n.$$

Similarly, by symmetry, we get $\phi(\eta^*) \leq \hat{\phi}_n(\hat{\eta}_n) + 3 \, \varepsilon_n$, so

$$\left| \hat{\phi}_n(\hat{\eta}_n) - \phi(\eta^*) \right| \ \leq \ 3 \, \varepsilon_n.$$

Since $\mathrm{CVaR}_\alpha(Y) = \phi(\eta^*)$ and $\widehat{\mathrm{CVaR}}_\alpha = \hat{\phi}_n(\hat{\eta}_n)$, we conclude

$$\left| \widehat{\mathrm{CVaR}}_\alpha - \mathrm{CVaR}_\alpha(Y) \right| \ \leq \ 3 \, \varepsilon_n \ = \ \mathcal{O}\!\left( \frac{M-m}{\alpha} \sqrt{\frac{\ln(1/\delta)}{n}} \right).$$

Finally, we absorb constant factors into a single $C$, yielding the stated bound. $\qquad\square$

**Theorem A.4** (Generalization Bound for Conditional CVaR Estimation). *Let $(X, Y)$ be distributed on $\mathcal{X} \times \mathbb{R}$, and let $\mathcal{G}$ be a class of measurable functions $g : \mathcal{X} \to \mathbb{R}$. Define the population Rockafellar–Uryasev (RU) risk of any predictor $g$ by*

$$R(g) \ := \ \mathbb{E}\!\left[ g(X) \ + \ \frac{1}{\alpha} \left( Y - g(X) \right)_+ \right],$$

*and let*

$$R^* \ = \ \inf_{g \in \mathcal{G}} \ R(g), \quad g^* \in \arg\min_{g \in \mathcal{G}} \ R(g).$$

*Given i.i.d. samples $\{(x_i, y_i)\}_{i=1}^n$, define the empirical RU risk*

$$\widehat{R}_n(g) \ := \ \frac{1}{n} \ \sum_{i=1}^n \left[ g(x_i) + \tfrac{1}{\alpha} \left( y_i - g(x_i) \right)_+ \right],$$

*and let*

$$\hat{g}_n \ \in \ \arg\min_{g \in \mathcal{G}} \ \widehat{R}_n(g).$$

*Suppose that, with probability at least $1 - \delta$,*

$$\sup_{g \in \mathcal{G}} \left| \widehat{R}_n(g) - R(g) \right| \ \leq \ \varepsilon_n,$$

*where $\varepsilon_n$ is a term that typically of the order $\mathcal{O}\!\left( \frac{1}{\alpha} \sqrt{\frac{\ln(1/\delta)}{n}} \right)$ under standard assumptions (boundedness, sub-Gaussian tails, etc.). Then on that event,*

$$R(\hat{g}_n) \ - \ R^* \ \leq \ 2 \, \varepsilon_n.$$

*Hence the learned predictor $\hat{g}_n$ achieves a CVaR-type risk within $2 \, \varepsilon_n$ of the best $g^* \in \mathcal{G}$, with high probability.*

*Proof.* **Step 1: Setup & Definitions.**

For each $g \in \mathcal{G}$, define the population RU risk

$$R(g) \ = \ \mathbb{E}\!\left[ g(X) + \tfrac{1}{\alpha} \left( Y - g(X) \right)_+ \right].$$

The empirical counterpart based on samples $(x_i, y_i)_{i=1}^n$ is

$$\widehat{R}_n(g) = \frac{1}{n} \sum_{i=1}^n \big[ g(x_i) + \tfrac{1}{\alpha}(y_i - g(x_i))_+ \big].$$

Let

$$\hat{g}_n \in \arg\min_{g \in \mathcal{G}} \widehat{R}_n(g), \quad g^* \in \arg\min_{g \in \mathcal{G}} R(g).$$

**Step 2: Decompose the Excess Risk.**

We want $R(\hat{g}_n) - R(g^*)$. Note that

$$R(\hat{g}_n) - R(g^*) = \underbrace{\big[R(\hat{g}_n) - \widehat{R}_n(\hat{g}_n)\big]}_{(A)} + \underbrace{\big[\widehat{R}_n(\hat{g}_n) - \widehat{R}_n(g^*)\big]}_{(B)} + \underbrace{\big[\widehat{R}_n(g^*) - R(g^*)\big]}_{(C)}.$$

Since $\hat{g}_n$ minimizes $\widehat{R}_n$, the middle term $(B) \leq 0$. Hence

$$R(\hat{g}_n) - R(g^*) \leq (A) + (C).$$

But

$$(A) = R(\hat{g}_n) - \widehat{R}_n(\hat{g}_n) \leq \sup_{g \in \mathcal{G}} \big| R(g) - \widehat{R}_n(g) \big|,$$

and similarly

$$(C) = \widehat{R}_n(g^*) - R(g^*) \leq \sup_{g \in \mathcal{G}} \big| \widehat{R}_n(g) - R(g) \big|.$$

Therefore,

$$R(\hat{g}_n) - R(g^*) \leq 2 \sup_{g \in \mathcal{G}} \big| \widehat{R}_n(g) - R(g) \big|.$$

**Step 3: Uniform Convergence Bound.**

By hypothesis (or by a standard Rademacher / VC argument), we have

$$\sup_{g \in \mathcal{G}} \big| \widehat{R}_n(g) - R(g) \big| \leq \varepsilon_n,$$

with probability $\geq 1 - \delta$, where $\varepsilon_n$ grows at a rate of $\mathcal{O}\big(\frac{1}{\alpha}\sqrt{\frac{\ln(1/\delta)}{n}}\big)$. Hence on that event:

$$R(\hat{g}_n) - R(g^*) \leq 2\varepsilon_n.$$

**Step 4: Why $\varepsilon_n$ Includes a Factor of $1/\alpha$.**

Observe that

$$\phi_\alpha(x, y; g) = g(x) + \frac{1}{\alpha}(y - g(x))_+.$$

Because it is scaled by $\frac{1}{\alpha}$, any standard concentration bound (*e.g.*, Hoeffding or Rademacher) for $\phi_\alpha$ incurs an extra factor of $1/\alpha$. Specifically:

- *Boundedness:* If $|g(x)| \leq G_{\max}$ and $|y| \leq Y_{\max}$, then $(y - g(x))_+ \leq |y - g(x)| \leq Y_{\max} + G_{\max}$. Hence $\phi_\alpha(x, y; g) \leq G_{\max} + \frac{1}{\alpha}(Y_{\max} + G_{\max})$.

- *Rademacher complexity or Hoeffding:* A uniform-convergence or covering-number argument yields a $\sqrt{\frac{\ln(1/\delta)}{n}}$ factor multiplied by the supremum of $|\phi_\alpha|$, which is $\leq \frac{C}{\alpha}$ for some constant $C$.

Thus $\varepsilon_n$ *necessarily* scales like $\frac{1}{\alpha}\sqrt{\frac{\ln(1/\delta)}{n}}$ (up to constants and possibly adding a $\mathfrak{R}_n(\mathcal{G})$ term if $\mathcal{G}$ is large).

$\square$

**Theorem A.5** (High-Dimensional Conditional CVaR Generalization Bound). *Let $(X, Y)$ be a random pair taking values in $\mathbb{R}^{d_x} \times \mathbb{R}^{d_y}$, and let $\alpha \in (0, 1)$ be fixed. Suppose we have:*

- *A scalar loss $\ell : \mathbb{R} \times \mathbb{R}^{d_y} \to \mathbb{R}$,*

- *A hypothesis class $\mathcal{G}$ of measurable functions $g : \mathbb{R}^{d_x} \to \mathbb{R}$,*

*and define the* Rockafellar–Uryasev (RU) risk *of any predictor $g \in \mathcal{G}$ by*

$$R(g) := \mathbb{E}\Big[g(X) + \frac{1}{\alpha}\Big(\ell\big(g(X), Y\big) - g(X)\Big)_+\Big].$$

*Let $R^* = \inf_{g \in \mathcal{G}} R(g)$, and choose $g^*$ such that $R(g^*) = R^*$. Given $n$ i.i.d. samples $\{(x_i, y_i)\}_{i=1}^n \subset \mathbb{R}^{d_x} \times \mathbb{R}^{d_y}$, define the* empirical *RU risk*

$$\widehat{R}_n(g) := \frac{1}{n}\sum_{i=1}^n \Big[g(x_i) + \frac{1}{\alpha}\big(\ell(g(x_i), y_i) - g(x_i)\big)_+\Big],$$

*and let $\hat{g}_n \in \arg\min_{g \in \mathcal{G}} \widehat{R}_n(g)$. Assume that with probability at least $1 - \delta$, we have a uniform-convergence bound*

$$\sup_{g \in \mathcal{G}}\Big|\widehat{R}_n(g) - R(g)\Big| \leq \varepsilon_n,$$

*where $\varepsilon_n$ scales as*

$$\varepsilon_n = \widetilde{\mathcal{O}}\Big(\frac{1}{\alpha}\sqrt{\frac{d_x + d_y}{n}}\Big),$$

*under suitable boundedness/sub-Gaussian assumptions on $(X, Y)$ and $\ell$. Then on that event,*

$$R(\hat{g}_n) - R^* \leq 2\varepsilon_n.$$

*Hence the learned predictor $\hat{g}_n$ achieves a CVaR-type risk within $2\varepsilon_n$ of the best $g^* \in \mathcal{G}$, with high probability.*

### A.3 GEN-DFL VS PRED-DFL

**Definition A.6.** Let $p(c|x)$ denote the true conditional distribution of $c$, and let $p_\theta(c|x)$ be the generative model. We define $Q_c$ to be the "worst $\alpha\%$ tail" representative for $c$ under $p(c|x)$ based on the target decision $w^\star$. Formally,

$$Q_c[\alpha] := \mathbb{E}[c \mid f(c, w^\star) \geq \text{VaR}_\alpha].$$

**Definition A.7.** Given the target decision $w^\star$, the decision $w^\star_{pred}$ found by Pred-DFL and the decision $w^\star_\theta$ found by `Gen-DFL`, we can define the regret of Pred-DFL formally as:

$$R_{\text{pred}}(x; \alpha) = f(Q_c[\alpha], w^\star_{\text{pred}}) - f(Q_c[\alpha], w^\star).$$

and the regret of `Gen-DFL` as:

$$R_\theta(x; \alpha) = \text{CVaR}_{p(c|x)}\Big[f(c, w^\star_\theta) - f(c, w^\star); \alpha\Big].$$

**Theorem A.8.** *Let $g : \mathcal{X} \to \mathcal{C}$ be the predictor in Pred-DFL. Assume the objective function $f(c, w)$ is Lipschitz continuous for any $c, w$. Then, there exists some constants $L_w, L_c, \kappa_1, \kappa_2, \kappa_3$ such that the following upper-bound holds for the aggregated regret gap $\mathbb{E}_x|\Delta R(x)|$:*

$$\mathbb{E}_x[\Delta R(x)] \leq \mathbb{E}_x\Big[\frac{2L_w}{\alpha}\big[\kappa_1\,\mathcal{W}(p_\theta, p) + \kappa_2\,\|\text{Bias}[g]\|\big] + (\frac{2L_w}{\alpha}\kappa_3 + 2L_c)\sqrt{\|\text{Var}[c \mid x]\|}$$
$$+ \text{CVaR}_{p(c|x)}[\|\text{Bias}[g(x)]\|; \alpha]\Big], \text{ where } \Delta R(x) = R_{\text{pred}}(x; \alpha) - R_{\text{gen}}(x; \alpha).$$

*Remark* A.9. Let $d_c$ and $d_x$ denote the dimension of $\mathcal{C}$ and $\mathcal{X}$, respectively. The bias term $\|\text{Bias}[g]\|$ of the predictor grows at a rate of $\mathcal{O}(\frac{1}{\alpha}\sqrt{(d_x + d_c)/n})$. This suggests that the smaller the $\alpha$ is, the harder for the predictor in the Pred-DFL to get an accurate estimation of $Q_c[\alpha]$.

*Remark* A.10. We may write $c = \bar{c} + \sigma\epsilon$, where $\bar{c} = \mathbb{E}_{p(c|x)}[c]$. Under some mild assumptions, such as $\epsilon$ being Gaussian, the variance term is of the order $\mathcal{O}(\sigma^2\sqrt{d_c})$.

*Proof.* **Step 1: Decomposition of the Regret**

$$\Delta R(x) = \text{CVaR}_{p(c|x)}[f(c, w_\theta^\star) - f(c, w^\star); \alpha] - \text{CVaR}_{p(c|x)}[f(c, w_{pred}^\star) - f(c, w^\star); \alpha]$$

$$= \text{CVaR}_{p(c|x)}[f(c, w_\theta^\star) - f(c, w_{pred}^\star); \alpha]$$

$$= \text{CVaR}_{p(c|x)}\big[\big[f(g(x), w_\theta^\star) - f(g(x), w_{pred}^\star)\big] + [f(c, w_\theta^\star) - f(g(x), w_\theta^\star)]$$

$$- [f(c, w_{pred}^\star) - f(g(x), w_{pred}^\star)]; \alpha\big]$$

$$\leq \big|\text{CVaR}_{p(c|x)}[f(g(x), w_\theta^\star) - f(g(x), w_{pred}^\star); \alpha]\big| + 2L_c\text{CVaR}_{p(c|x)}[\|c - g(x)\|; \alpha]$$

$$= \big|\text{CVaR}_{p(c|x)}[f(g(x), w_\theta^\star) - f(g(x), w_{pred}^\star); \alpha]\big| + 2L_c\text{CVaR}_{p(c|x)}[\|c - Q_c[\alpha] + Q_c[\alpha] - g(x)\|; \alpha]$$

$$\leq \big|\text{CVaR}_{p(c|x)}[f(g(x), w_\theta^\star) - f(g(x), w_{pred}^\star); \alpha]\big| + 2L_c\big|\text{CVaR}_{p(c|x)}[\|c - Q_c[\alpha]\|; \alpha]$$

$$+ \text{CVaR}_{p(c|x)}[\|\text{Bias}[g]\|; \alpha]\big|$$

$$\leq \big|\text{CVaR}_{p(c|x)}[f(g(x), w_\theta^\star) - f(g(x), w_{pred}^\star); \alpha]\big| + 2L_c\big|\sqrt{d\|Var[c \mid x]\|} + \text{CVaR}_{p(c|x)}[\|\text{Bias}[g]\|; \alpha]\big|$$

where we used the fact $f(c, w_\theta^\star) = f(g(x), w_\theta^\star) + [f(c, w_\theta^\star) - f(g(x), w_\theta^\star)]$ and $f(c, w_{pred}^\star) = f(g(x), w_{pred}^\star) + [f(c, w_{pred}^\star) - f(g(x), w_{pred}^\star)]$.

**Step 2: Bounding** $\Delta_{\text{Term}} = \big|\textbf{CVaR}_{p(c|x)}[f(g(x), w_\theta^\star) - f(g(x), w_{pred}^\star); \alpha]\big|$

Now, we need to bound the $\Delta_{\text{Term}} = \big|\text{CVaR}_{p(c|x)}[f(g(x), w_\theta^\star) - f(g(x), w_{pred}^\star); \alpha]\big|$ term.

By assumption, for any fixed $c_0$, the map $w \mapsto f(c_0, w)$ is $L_w$-Lipschitz in $w$. Equivalently,

$$\big|f(c_0, w_1) - f(c_0, w_2)\big| \leq L_w\|w_1 - w_2\|.$$

Applying this specifically at $c_0 = g(x)$, we get:

$$\big|f(g(x), w_\theta^\star) - f(g(x), w_{\text{pred}}^\star)\big| \leq L_w\|w_\theta^\star - w_{\text{pred}}^\star\|.$$

Since $\text{CVaR}_p[\cdot]$ is merely an expectation that does not affect the integrand here (it does not depend on $c$ anymore), we have

$$\Delta_{\text{Term}} \leq L_w\|w_\theta^\star - w_{\text{pred}}^\star\|.$$

**Step 3: Bounding** $\|w_\theta^\star - w_{\text{pred}}^\star\|$

First, we define the following auxiliary (aggregate objectives) functions for both `Gen-DFL` and `Pred-DFL`,

$$\text{Gen-DFL: } J_{\text{gen}}(w) = \text{CVaR}_{p_\theta}[f(c, w); \alpha], \quad \text{Pred-DFL: } J_{\text{pred}}(w) = f(g(x), w).$$

So

$$w_\theta^\star = \arg\min_w J_{\text{gen}}(w), \quad w_{\text{pred}}^\star = \arg\min_w J_{\text{pred}}(w).$$

Next, let's define

$$\Delta(w) = J_{\text{gen}}(w) - J_{\text{pred}}(w) = \text{CVaR}_{p_\theta}[f(c, w); \alpha] - f(g(x), w).$$

We take a uniform bound over $w$:

$$T = \sup_w |\Delta(w)|.$$

We will then show that,

$$T \leq \kappa_1\|p_\theta - p\| + \kappa_2\sqrt{\|\text{Var}[c \mid x]\|} + \kappa_3\|\text{Bias}[g]\|.$$

**Step 4: Bounding** $T$

By definition,

$$T := \sup_w \Big|\text{CVaR}_{p_\theta}[f(c, w); \alpha] - f(g(x), w)\Big|.$$

To relate this to the *true* distribution $p$ and $Q_c[\alpha] = \mathbb{E}_{c \sim p}[c]$, we can do the following decomposition:

$$\text{CVaR}_{p_\theta}[f(c, w); \alpha] - f(g(x), w) = \Big(\text{CVaR}_{p_\theta}[f(c, w); \alpha] - \text{CVaR}_p[f(c, w)]\Big)$$
$$+ \Big(\text{CVaR}_p[f(c, w); \alpha] - f(Q_c[\alpha], w)\Big) + \Big(f(Q_c[\alpha], w) - f(g(x), w)\Big).$$

Hence, if we set

$$T = \sup_w \big|(A) + (B) + (C)\big|,$$

then by triangle inequality:

$$T \leq \underbrace{\sup_w \big|(A)\big|}_{T_1} + \underbrace{\sup_w \big|(B)\big|}_{T_2} + \underbrace{\sup_w \big|(C)\big|}_{T_3}.$$

We now bound each piece $T_1, T_2, T_3$ separately.

First, we can see that

$$T_1 = \sup_w \big|\text{CVaR}_{p_\theta}[f(c, w); \alpha] - \text{CVaR}_p[f(c, w)]\big| \leq \kappa_1 \, \mathcal{W}(p_\theta, p),$$

where $\kappa_1$ depends on the Lipschitz constant of $f$ in $c$.

Next, by taking the Taylor expansion, we have

$$f(c, w) = f(g(x), w) + \nabla_c f(g(x), w)^T (c - g(x)) + \frac{1}{2}(c - g(x))^T \nabla_c^2 f(g(x), w)(c - g(x)) + \mathcal{O}(||c - g(x)||^2)$$

After taking the CVaR expectation, we see that

$$T_2 = \sup_w \big|\text{CVaR}_p[f(c, w); \alpha] - f(g(x), w)\big| \leq \kappa_2 \sqrt{||\text{Var}[c \mid x]||},$$

where $\kappa_2$ incorporates the Lipschitz constant.

Finally, for $T_3$, assuming that $f(\cdot, w)$ is Lipschitz in $c$, then

$$T_3 = \sup_w \big|f(Q_c[\alpha], w) - f(g(x), w)\big| \leq L_c \, \|Q_c[\alpha] - g(x)\| \leq L_c \, \|\text{Bias}[g]\|.$$

Hence,

$$T_3 \leq \kappa_3 \, \|\text{Bias}[g]\|.$$

**Combining all the steps.**

Collecting $T_1, T_2, T_3$:

$$T = \sup_w \big|\text{CVaR}_{p_\theta}[f(c, w); \alpha] - f(g(x), w)\big| \leq T_1 + T_2 + T_3$$

$$\leq \kappa_1 \, \mathcal{W}(p_\theta, p) + \kappa_2 \sqrt{||\text{Var}[c \mid x]||} + \kappa_3 \, \|\text{Bias}[g]\|.$$

Thus,

$$\boxed{T \leq \kappa_1 \, \mathcal{W}(p_\theta, p) + \kappa_2 \sqrt{||\text{Var}[c \mid x]||} + \kappa_3 \, \|\text{Bias}[g]\|.}$$

**Strong Convexity in $w$ Yields Solution Stability.**

Assume $J_{\text{gen}}(\cdot)$ and $J_{\text{pred}}(\cdot)$ are $\alpha$-strongly convex in $w$. Then,

$$\big\|w_\theta^\star - w_{\text{pred}}^\star\big\| \leq \frac{2}{\alpha} \sup_w \big|\Delta(w)\big| = \frac{2}{\alpha} T.$$

Therefore,

$$\big\|w_\theta^\star - w_{\text{pred}}^\star\big\| \leq \frac{2}{\alpha}\Big(\kappa_1 \, \mathcal{W}(p_\theta, p) + \kappa_2 \sqrt{||\text{Var}[c \mid x]||} + \kappa_3 \, \|\text{Bias}[g]\|\Big).$$

**Combining all the steps**

$$\Delta_{\text{Term}} = \left| \text{CVaR}_p[f(g(x), w_\theta^\star) - f(g(x), w_{\text{pred}}^\star); \alpha] \right| \leq L_w \left\| w_\theta^\star - w_{\text{pred}}^\star \right\| \leq L_w \frac{2}{\alpha} T,$$

Therefore,

$$\Delta_{\text{Term}} \leq L_w \cdot \frac{2}{\alpha} \left[ \kappa_1 \, \mathcal{W}(p_\theta, p) + \kappa_2 \sqrt{\|\text{Var}[c \mid x]\|} + \kappa_3 \|\text{Bias}[g]\| \right].$$

Finally, we get,

$$\mathbb{E}_x |\Delta R(x)| \leq \mathbb{E}_x \Big[ L_w \cdot \frac{2}{\alpha} \left[ \kappa_1 \, \mathcal{W}(p_\theta, p) + \kappa_2 \sqrt{\|\text{Var}[c \mid x]\|} + \kappa_3 \|\text{Bias}[g]\| \right]$$
$$+ 2L_c \Big| \sqrt{d \|Var[c \mid x]\|} + \text{CVaR}_{p(c|x)} [\|\text{Bias}[g(x)]\|]; \alpha \Big| \Big].$$

Moreover, by Theorem A.5 that we developed earlier, we can see the bias term $\|\text{Bias}[g]\|$ grows at a rate of $\mathcal{O}(\frac{1}{\alpha} \sqrt{(d_x + d_c)/n})$

$\square$

## B EXPERIMENTAL SETUPS

### B.1 SYNTHETIC: PORTFOLIO OPTIMIZATION

In the Portfolio experiment, we generate the synthetic data as follows:

$$x_i \sim \mathcal{N}(0, I^{d_x}),$$
$$\mathbf{B}_{ij} \sim \text{Bernoulli}(0.5),$$
$$L_{ij} \sim \text{Uniform}[-0.0025\sigma, 0.0025\sigma]$$
$$\epsilon \sim \mathcal{N}(0, I^{d_c})$$
$$\bar{c}_{ij} = \left( \frac{0.05}{\sqrt{p}} \mathbf{B} x_i + 0.1 \right)^{deg} + Lf + 0.01\sigma\epsilon,$$

where $d_x, d_c$ are the dimensionality of the input features $x$ and the cost vector $c$. The polynomial degree reflects the level of non-linearity between the feature and the price vector. In Portfolio, $c$ represents the asset prices and the dimension of $c_i$ is the number of assets.

The non-linear, risk-sensitive optimization problem in Portfolio Management is then formulated as,

$$w^\star(x) := \min_w \text{CVaR}_{p(c|x)}[-c^T w + w^T \Sigma w; \alpha]$$
$$\text{s.t. } w \in [0, 1]^n, \; \mathbf{1}^T w \leq 1, \tag{10}$$

where $\Sigma = LL^T + (0.01\sigma)^2 I$ is the covariance over the asset prices $c$, and the quadratic term $w^T \Sigma w$ reflects the amount of risk.

### B.2 SYNTHETIC: FRACTIONAL KNAPSACK

In the Knapsack experiment, we generate the synthetic data as follows:

$$x_i \sim \mathcal{N}(0, I^{d_x}),$$
$$\mathbf{B}_{ij} \sim \text{Bernoulli}(0.5),$$
$$L_{ij} \sim \text{Uniform}[-0.0025\sigma, 0.0025\sigma]$$
$$\epsilon \sim \mathcal{N}(0, I^{d_c})$$
$$\bar{c}_{ij} = \left( \frac{0.05}{\sqrt{p}} \mathbf{B} x_i + 0.1 \right)^{deg} + Lf + 0.01\sigma\epsilon,$$

where $d_x, d_c$ are the dimensionality of the input features $x$ and the cost vector $c$.

The optimization problem in Knapsack is formulated as:

$$w^\star(x) := \min_w \mathrm{CVaR}_{p(c|x)}[-c^T w; \alpha] \tag{11}$$
$$\text{s.t. } w \in [0, 1]^n, p^T w \leq \mathbf{B},$$

where $p \in \mathbb{R}^n$ and $\mathbf{B} > 0$ represent the capacity and weight vector, respectively.

### B.3 SYNTHETIC: SHORTEST-PATH

In the Shortest-Path experiment, we generate the synthetic data as follows:

$$x_i \sim \mathcal{N}(0, I^{d_x}),$$
$$\mathbf{B}_{ij} \sim \mathrm{Bernoulli}(0.5),$$
$$\epsilon_{ij} \sim \mathrm{Uniform}[0.5, 1.5]$$
$$\bar{c}_{ij} = \left[ \frac{1}{3.5^{deg}} \left( \frac{1}{\sqrt{p}} \mathbf{B} x_i + 3 \right)^{deg} + 1 \right] \cdot \epsilon_i^j,$$

where $d_x, d_c$ are the dimensionality of the input features $x$ and the cost vector $c$. The polynomial degree reflects the level of non-linearity between the feature and the price vector.

The optimization problem in Shortest-Path is formulated as:

$$w^\star(x) := \min_w \mathrm{CVaR}_{p(c|x)}[c^T w; \alpha] \tag{12}$$
$$\text{s.t. } w \in [0, 1]^n,$$

where $c^T w$ represents the cost of the selected path, and the cost vector $c_i^j$ is defined as follows:

$$c_i^j = \left[ \frac{1}{3.5^{deg}} \left( \frac{1}{\sqrt{p}} \mathbf{B} x_i + 3 \right)^{deg} + 1 \right] \cdot \epsilon_i^j,$$

where $\mathbf{B}$ is a random matrix, and $\epsilon_i^j$ is the noise component.

The features $x_i \in \mathbb{R}^{d_x}$ follow a standard multivariate Gaussian distribution, and the uncertain coefficients $c_i^j$ exist only on the objective function, meaning that the weights of the items remain fixed throughout the dataset. The parameters include the dimension of resources $k$, the number of items $m$, and the noise width.

### B.4 REAL DATASET: ENERGY-COST AWARE SCHEDULING PROBLEM

In this task, we consider a demand response program in which an operator schedules electricity consumption $p_t$ over a time horizon $t \in \Omega_t$. The objective is to minimize the total cost of electricity while adhering to operational constraints. The electricity price for each time step is denoted by $\pi_t$, which is not known in advance. However, the operator can schedule the electricity consumption $p_t$ within a specified lower bound $P_t$ and upper bound $\overline{P}_t$. Additionally, the total consumption for the day, denoted as $P_t^{\mathrm{sch}}$, must remain constant. This assumes flexibility in shifting electricity demand across time steps, provided the total demand is met.

The optimization problem, assuming perfect information about prices $\pi_t$, can be formulated as:

$$\min_{p_t} \mathrm{CVaR}_\pi \left[ \sum_{t \in \Omega_t} \pi_t p_t; \alpha \right],$$

subject to the constraints:

$$P_t \leq p_t \leq \overline{P}_t, \quad \forall t,$$
$$\sum_{t \in \Omega_t} p_t = \sum_{t \in \Omega_t} P_t^{\mathrm{sch}}.$$

---

**Algorithm 1** Learning Algorithm for `Gen-DFL`

---

**Input:** Dataset $\mathcal{D} = \{(x_i, c_i)\}_{i=1}^{N}$, CGM $p_\theta(c|x)$, learning rate $\eta$, regularization ratio $\gamma$, sampling size $K$, risk-level $\alpha$, a proxy model $q(c|x)$ trained on $\mathcal{D}$.

**while** not converged **do**

$\quad \{c_k\}_{k=1}^{K} \sim p_\theta(c|x); \; N_k \leftarrow$ number of $c_k$ that satisfy $\mathbb{1}\{f(c_k, w) \geq \text{VaR}_\alpha\}$

$\quad w_\theta^\star \leftarrow \arg\min_{w} \frac{1}{N_k} \sum_{k=1}^{K} f(c_k, w) \mathbb{1}\{f(c_k, w) \geq \text{VaR}_\alpha\};$

$\quad \ell(\theta; q, \alpha) \leftarrow \frac{1}{n} \sum_{i=1}^{n} \text{Regret}_{\theta, q}(x_i; \alpha) + \gamma \cdot \ell_{\text{gen}}(\theta);$

$\quad \theta \leftarrow \theta - \eta \cdot \partial\ell / \partial\theta;$

**end while**

---

Here, $P_t \leq p_t \leq \overline{P}_t$ ensures the consumption at each time step is within the allowed bounds, while the equality constraint guarantees that the total electricity consumption remains fixed across the time horizon.

This setup reflects the practical challenges of demand-side electricity management, where prices are uncertain, and demand shifting across time steps provides opportunities for cost reduction while maintaining overall consumption levels. The problem serves as a testbed for evaluating optimization approaches under uncertain electricity prices and operational constraints.

### B.5 COVID RESOURCE ALLOCATION

The COVID-19 pandemic has highlighted the challenges policymakers and epidemiologists face in planning for surges in medical resource demand, such as hospital beds. As the number of infected patients increases, accurate forecasts of hospitalizations become critical for effective resource allocation. To achieve this, epidemiological models based on Ordinary Differential Equations (ODEs) are often employed to capture and forecast the dynamics of infectious disease outbreaks. These forecasts are then used as guidance for planning future resource allocation.

In this task, we focus on the optimization problem of hospital bed preparation during a pandemic, a critical task for ensuring adequate medical infrastructure. Specifically, the goal is to decide how many hospital beds $a \in \mathbb{R}^7$ to allocate for the next seven days based on the forecasted number of hospitalized patients $y \in \mathbb{R}^7$. The optimization objective combines linear and quadratic costs to account for both over-preparation ($[a_i - y_i]_+$) and under-preparation ($[y_i - a_i]_+$) of hospital beds, ensuring both efficiency and safety in resource allocation.

The optimization problem is formulated as:

$$\min_{a \in \mathbb{R}^7} \sum_{i=1}^{7} c_b[a_i - y_i]_+ + c_h[y_i - a_i]_+ + q_b([y_i - a_i]_+)^2 + q_h([a_i - y_i]_+)^2,$$

where $c_b$ and $c_h$ represent the linear cost coefficients for over- and under-preparation, while $q_b$ and $q_h$ represent the quadratic penalty coefficients for the same. These terms reflect the trade-offs between allocating too many beds, which leads to wasted resources, and too few beds, which risks inadequate care for patients.

This formulation integrates ODE-based forecasts to guide decision-making, enabling a data-driven approach to resource planning under uncertainty. The problem is designed to balance competing objectives effectively, ensuring sufficient resources while minimizing waste during critical periods of high demand.

## C IMPLEMENTATION

### C.1 ALGORITHM

The details of `Gen-DFL` implementation can be found in Algorithm 1 .

| Task | Method | Learning Rate | Variance $\sigma$ | Dimension $d$ | Training Size | $\beta$ |
|---|---|---|---|---|---|---|
| | Pairwise | $10^{-3}$ | 20 | 50 | 320 | - |
| | Listwise | $10^{-3}$ | 20 | 50 | 320 | - |
| | NCE | $10^{-3}$ | 20 | 50 | 320 | - |
| **Portfolio** | MAP | $10^{-3}$ | 20 | 50 | 320 | - |
| | SPO | $10^{-3}$ | 20 | 50 | 320 | - |
| | MSE (PTO) | $10^{-3}$ | 20 | 50 | 320 | - |
| | Gen-DFL | $10^{-3}$ | 20 | 50 | 320 | 10.0 |
| | Pairwise | $10^{-3}$ | 20 | 50 | 320 | - |
| | Listwise | $10^{-3}$ | 20 | 50 | 320 | - |
| | NCE | $10^{-3}$ | 20 | 50 | 320 | - |
| **Knapsack** | MAP | $10^{-3}$ | 20 | 50 | 320 | - |
| | SPO | $10^{-3}$ | 20 | 50 | 320 | - |
| | MSE (PTO) | $10^{-3}$ | 20 | 50 | 320 | - |
| | Gen-DFL | $10^{-3}$ | 20 | 50 | 320 | 10.0 |
| | Pairwise | $10^{-1}$ | 5 | 25 | 320 | - |
| | Listwise | $10^{-1}$ | 5 | 25 | 320 | - |
| | NCE | $10^{-1}$ | 5 | 25 | 320 | - |
| **Shortest-Path** | MAP | $10^{-1}$ | 5 | 25 | 320 | - |
| | SPO | $10^{-1}$ | 5 | 25 | 320 | - |
| | MSE (PTO) | $10^{-1}$ | 5 | 25 | 320 | - |
| | Gen-DFL | $10^{-3}$ | 20 | 50 | 320 | 10.0 |

Table 2: Hyperparameters and Problem Configurations

## C.2 HYPERPARAMETER CONFIGURATIONS

Table 2 summarizes the hyperparameter settings and problem configurations across different tasks and baselines. For all methods, we maintain a consistent number of training samples ($n = 320$) and input dimensionality ($d = 50$ for Portfolio and Knapsack, $d = 25$ for Shortest-Path) to ensure a fair comparison. The learning rates vary across tasks, with a higher value (0.1) used for the Shortest-Path problem, reflecting its different optimization landscape. The noise scale $\sigma$ remains fixed at 20 for Portfolio and Knapsack, while a lower value ($\sigma = 5$) is used for Shortest-Path to account for its different problem structure.

For Gen-DFL, we introduce an additional DFL loss weight $\beta$ which controls the balance between the decision-focused objective and the negative log-likelihood (NLL) regularization, so that

$$\ell_{\texttt{Gen-DFL}}(\theta; q, \alpha) := \beta \cdot \mathbb{E}_x[\text{Regret}_{\theta,q}(x; \alpha)] + \gamma \cdot \ell_{\text{gen}}(\theta).$$

Unlike baseline Pred-DFL models, which optimize directly over point estimates, Gen-DFL leverages generative modeling and requires careful tuning of $\beta, \gamma$ to ensure stable training. The uniformity in hyperparameter selection across methods helps isolate the impact of different learning paradigms.

## D COMPUTATIONAL COMPLEXITY

Regarding training cost, our method scales linearly with the number of Monte Carlo samples used in the sample average approximation (SAA). Our experiments on a consumer-grade Apple M2 CPU show: with 200 generated samples per decision, Gen-DFL is roughly five times slower to train than Pred-DFL, and with 800 samples it can be up to twenty times slower. In the portfolio experiment, for example, training Pred-DFL takes about 8 minutes, whereas Gen-DFL with 200 generated samples takes roughly 40 minutes. This overhead stems from our desire to model the full conditional distribution $p(c|x)$ expressively; each additional sample improves the approximation of the CVaR but also increases runtime. We emphasize that this computational overhead does not pose a significant drawback since inference typically relies on pre-trained models rather than real-time training.

# E STATISTICAL ANALYSIS

| Task | Setting | p-value (Gen-DFL vs 2Stage) | p-value (Gen-DFL vs SPO+) |
|---|---|---|---|
| Portfolio | Deg-2 | < 0.0001 | < 0.0001 |
| Portfolio | Deg-4 | < 0.0001 | < 0.0001 |
| Portfolio | Deg-6 | < 0.0001 | < 0.0001 |
| Portfolio | Deg-8 | < 0.0001 | < 0.005 |
| Knapsack | Deg-2 | 0.79 | 0.61 |
| Knapsack | Deg-4 | 0.80 | 0.43 |
| Knapsack | Deg-6 | 0.59 | 0.95 |
| Knapsack | Deg-8 | 0.83 | 0.77 |
| Shortest Path | Deg-2 | < 0.0001 | 0.085 |
| Shortest Path | Deg-4 | < 0.0001 | 0.038 |
| Shortest Path | Deg-6 | < 0.0001 | 0.046 |
| Shortest Path | Deg-8 | < 0.0001 | 0.006 |
| Energy | – | 0.003 | 0.004 |

Table 3: Comparison: Gen-DFL vs 2Stage (PTO) and SPO+

