# OpenReview forum: "Gen-DFL: Decision-Focused Generative Learning for Robust Decision Making"
_ICLR.cc/2026/Conference — ICLR 2026 Poster_

### Official Review · Reviewer_Uiro · 2025-10-29

**Soundness:** 3
**Presentation:** 3
**Contribution:** 3
**Rating:** 4
**Confidence:** 5

**Summary:**

The paper combines the advantages of Decision-focused learning (DFL) and robust optimization (RO) to propose a decision-making framework robust to model uncertainty. The authors introduce a method that uses the conditional distribution learned by generative models instead of uncertainty sets to protect the decision-making against tail regions of the distribution. This results in optimizing a Conditional Value-at-Risk objective. The paper provides theoretical results on the loss difference using an approximation of the conditional distribution and regret gap between standard DFL and their method. They validate their methods on an energy-cost aware scheduling problem and the COVID-19 resource allocation problem, along with three synthetic experiments.

**Strengths:**

The paper proposes a method that addresses the faults of RO and DFL. I believe the introduction, related works, and preliminaries provide sufficient context for the paper. The method was robustly validated in experiments with a wide range of data sets and ablation studies. The method was compared against many baselines in RO and DFL. The empirical results make a strong and convincing case for using this method.

**Weaknesses:**

Theorem 5.4, which is one of the two main theoretical contributions of the paper, studies the upper bound of $\mathbb{E}_x\left[|\Delta R(x)|\right]$ to characterize the factors that influence the performance gap between standard DFL and Gen-DFL. However, the authors analyze these factors to highlight the failure modes of Pred-DFL, which is reflected by $\mathbb{E}_x\left[\Delta R(x)\right]$ instead of  $\mathbb{E}_x\left[|\Delta R(x)|\right]$. While the analysis of the performance gap is interesting, it doesn’t strengthen the case made for Gen-DFL.

Additionally, it seems like the performance gap, as defined by the authors, should also have a distribution distance term between $p(c\mid x)$ and $q(c\mid x)$ (similar to the result in Theorem 5.1). It seems odd that the miscalibration of the generative model has no effect on the performance gap.

If a satisfactory upper bound and lower bound for $\mathbb{E}_x\left[\Delta R(x)\right]$ is provided, I would be willing to increase my score.

Minor weaknesses
* The theorem statement of Theorem 5.4 isn’t the same as Theorem A.8.
* In lines 344-345, the authors argue that the estimation error of the predictor in Pred-DFL grows at a certain rate, but I don’t think this is immediately obvious. It would’ve been nice if there were some explanation around that statement or a reference to the original result.

**Questions:**

* Why aren’t the RO baselines included in Table 1?
* What data sets are used to produce Figure 3-7?
* Why is the $\beta$ hyperparameter introduced in the experiments section? Why can’t $\gamma$ be varied in experiments instead of $\beta$?
* What is this “auxiliary model” in line 259? Are there two models in Gen-DFL?

---

> ### Author Response · Authors · 2025-11-23
>
> We thank the reviewer for the insightful feedback. First, we apologize for the confusion caused by our writing as we had to move a lot of technical details which can address your questions to the appendix due to space limitation. We will submit a revised manuscript which adds all the missing references to our appendix, and we will consider moving some essential results from the appendix to the main text if our manuscript is accepted and given extra page. We will now address your concern regarding our Theorem in the following:
>
> > The bound of $\mathbb{E}[\Delta R(x)]$
>
> We argue that it is equivalent to the one bounding $\mathbb{E}[|\Delta R(x)|]$. This is because the absolute sign takes no effect in the derivation from line 974 to 977 as shown below.
>
> $\Delta R(x) = CVaR_{p(c|x)}[f(c, w_\theta^\star) - f(c, w^\star);\alpha] - CVaR_{p(c|x)}[f(c, w_{pred}^\star) - f(c, w^\star);\alpha] $
>
> $= CVaR_{p(c|x)}[f(c, w_\theta^\star) - f(c, w_{pred}^\star);\alpha]$
>
> $= CVaR_{p(c|x)}\left[[f(g(x), w_\theta^\star) - f(g(x), w_{pred}^\star)] + \left[ f(c, w_\theta^\star) - f(g(x), w_\theta^\star) \right] - [ f(c, w_{pred}^\star) - f(g(x), w_{pred}^\star) ] ;\alpha \right]$
>
> using the fact that $f(c, w_\theta^\star) = f(g(x), w_\theta^\star) + [f(c, w_\theta^\star) - f(g(x), w_\theta^\star)]$ and $f(c, w_{pred}^\star) = f(g(x), w_{pred}^\star) + [f(c, w_{pred}^\star) - f(g(x), w_{pred}^\star)]$.
>
> And the rest of the proof will follow the same steps. We will remove the absolute sign in our revised manuscript. We thank the reviewer for pointing this out.
>
> > Theorem 5.4/A.8 and distribution distance.
>
> We respectfully clarify that the results in Theorem 5.4 and Theorem A.8 are equivalent, with some slight differences in terms of how each term is arranged.
> The distance between $p_\theta(c|x)$ and $p(c|x)$ is characterized by the Wasserstein distance $W(p_\theta, p)$, which is the first term in the bound. We apologize if the writing is ambiguous, and we will make sure that all the terms are arranged in the same way in our revised manuscript.
>
> > In lines 344-345, the authors argue that the estimation error of the predictor in Pred-DFL grows at a certain rate
>
> This is an immediate result following our conclusion in Theorem A.5 in the appendix. We had to move the theorem to the appendix due to space limits, and we will update the manuscript and include a reference here in the main text.
>
> > Why aren’t the RO baselines included in Table 1
>
> We will address this in our updated manuscript this week.
>
> > Figure 3-7?
>
> These results are gathered using the portfolio data.
>
> > Why is the $\beta$ hyperparameter introduced in the experiments section? Why can’t $\gamma$ be varied in experiments instead of $\beta$?
>
> We control the weight $\beta$ of the decision loss as we want to analyze how the decision quality changes if we put different mass of decision loss onto a vanilla generative model loss $\gamma=1$. In practice, one can play around with both hyperparameters so we included $\gamma$ at the beginning, but we later found that tweeking the weight $\beta$ of the decision loss can already achieve what we’d like to analyze.
>
> This also depends on how one interprets the relationship between the decision loss and the likelihood loss. One can say that the decision loss is the main focus of the optimization and the likelihood loss serves as a regulation so that the model won't deviate from the empirical data distribution too much or focus solely on the tail end, and by convention the weight of the regularizer should be the hyperparameter ($\gamma$ in this case); or the other way around. Since the decision loss is the main focus here, we introduced the weight of the likelihood loss as the hyperparam.
>
> We will consider removing $\gamma$ as it is set to be 1 anyway and introducing $\beta$ in the method section rather than in the experimental section if that makes the writing more easy to follow.
>
> > What is this “auxiliary model”?
>
> The auxiliary model $q(c|x)$ is the approximation of $p(c|x)$ as we assume that $p(c|x)$ is not given during the DFL training stage. Before the training stage of Gen-DFL starts, an auxiliary model $q(c|x)$ will first be trained to approximate empirical data distribution, for which it is usually a near perfect fit under the dataset settings considered in this manuscript. This is also detailed in our Algorithm 1 from line 1188 in the appendix. Again, we moved the algorithm box to the appendix due to space limit and we will include in the main text with additional explanation if our manuscript is accepted and given an extra page.
>
> We will update the manuscript to address the above issues you pointed out. We thank the reviewer again for your valuable comments and questions. If all your questions and concerns are answered, we would hope you can reconsider your assessment and increase the score if you think our updated manuscript is improved. We remain ready to address any further concerns if needed. Many thanks!

---

> ### Author Response · Authors · 2025-11-26
>
> Dear Reviewer,
>
> as the discussion period is nearing its end, we wanted to ask if there are any follow-up points we can clarify.
>
> We believe we have responded to all questions and concerns raised. If there are no further points of clarification regarding the manuscript, we kindly ask that reviewer Uiro consider increasing their score to reflect the improvements and clarifications we have provided. We are happy to continue to engage in discussion and answer any questions!

---

### Official Review · Reviewer_eGUr · 2025-10-31

**Soundness:** 3
**Presentation:** 3
**Contribution:** 3
**Rating:** 6
**Confidence:** 3

**Summary:**

The paper tackles the problem of making decisions via predictive models. Specifically, this paper focuses on the end-to-end regime of decision focused learning (DFL) which has been explored in recent years. For DFL, the model often takes the form of bilevel structure with a predictive model (for the cost vector) and an inner minimization of a downstream expected objective. This paper defines gen-DFL which instead of a predictor uses a generative model that tries to predict a distribution rather than a point prediction of the cost vector. This results in improvements in high dimensional settings over baselines and the authors provide theoretical bounds on performance.

**Strengths:**

It has clear empirical improvements and clear theory on generalization bounds and provides an interesting alternative method to the other formulations of DFL.

**Weaknesses:**

The justification of DFL versus point-wise robust methods is not theoretically clear. First, while they provide gaps between the point-prediction methods of pred-DFL and gen-DFL, first it is unclear what the \Delta R term really means. The differing definitions of regret for gen-DFL and pred-DFL make sense as one is with a regret realized by a singular cost value and the other is over a distribution of realizations. To then take the difference between these two terms is what seems a little strange. Later it seems that the definition of \Delta R in appendix A.8 uses the same CvaR regret for both but doesn’t use the defined pred-DFL regret. Thus, the \Delta R term is confusing and some clarification would help here.

Furthermore, the purpose of this bound is a little unclear. If the motive is to demonstrate theoretical improvements of the gen-DFL method over pred-DFL type methods, why is it an upper-bound on the absolute residual gap? Shouldn’t it instead be a lower-bound type result on the gap without absolute difference such that the regret of pred-DFL methods is at least some quantity larger than the regret of gen-DFL?
So while the upper-bound is valid, it doesn’t necessarily distinguish the two methods, and perhaps a corresponding lower-bound would help.

Lastly, while there are many solid synthetic experiments, there are only 2 real-world data ones, for which one of them diff-DRO outperforms (admittedly by a small amount). It would be nice to include another such experiment in which the empirical performance of gen-DFL on real data is demonstrated. Additionally the results on these real world experiments seems lacking.

**Questions:**

Could the computational overhead of gen-DFL be problematic under settings that require retraining under distribution shift, perhaps? Is there an equivalence in solutions between the "right size" of an uncertainty set for pred-DFL with robust optimization and the full tail distribution of gen-DFL?

---

> ### Author Response · Authors · 2025-12-01
>
> We thank the reviewer for your insightful feedback. We will now address your questions as below:
>
> > The bound of $\mathbb{E}[\Delta R(x)]$
>
> First, we agree that a lower bound on the difference $(R_{pred} - R_{gen})$ would theoretically prove that Gen-DFL is always better (which is probably not the case and the proof is most likely intractable). However, we respectfully clarify that the primary purpose of **Theorem 5.4** is to understand the failure modes of Pred-DFL by decomposing the error terms in comparison to Gen-DFL.
>
> Here are more details about the implications of the upper-bound in Theorem 5.4.
>
> 1.  $$\kappa_2 \sqrt{\|Var[c|x]\|} + \kappa_3 \|Bias[g(x)]\|$$ quantifies how the error of Pred-DFL scales. As noted in **Remark A.9** (Appendix A.8), the bias term $\|Bias[g]\|$ for point estimators typically grows at a rate of $\mathcal{O}(\frac{1}{\alpha}\sqrt{(d_x + d_c)/n})$.
>
> This explains why Pred-DFL degrades rapidly in high-dimensional ($d_c \uparrow$) or high-variance settings (demonstrated by our experiments in **Figure 4**), as the point estimate $g(x)$ becomes an increasingly poor proxy for the risk-adjusted parameter $Q_c[\alpha]$.
>
> 2. In contrast, the error contribution from Gen-DFL is bounded primarily by the Wasserstein distance $\kappa_1 \mathcal{W}(p_\theta, p)$. This shows that Gen-DFL's performance depends on the quality of the generative distribution matching, rather than the variance of the data itself.
>
> Now, to your questions:
>
> > Could the computational overhead of gen-DFL be problematic under settings that require retraining under distribution shift, perhaps?
>
> We acknowledge that Gen-DFL incurs higher training costs than point-estimate methods due to the generative modeling component. However, the data distributions we often encounter in decision-making tasks are fairly simple compared to those in image/text domains so the generative models are not hard to learn. Considering the benefit it brings in risk-sensitive tasks, we would regard the additional training costs to be trivial if it’s just several more hours of training.
>
> > Is there an equivalence in solutions between the "right size" of an uncertainty set for pred-DFL with robust optimization and the full tail distribution of gen-DFL?
>
> This is a good question. First, we’d like to clarify that,  for any arbitrary size uncertainty set, Pred-DFL with robust optimization only protects against the worst case in that specific uncertainty set, while Gen-DFL protects against the all the risky cases beyond certain risk threshold. Second, even when the ‘right-size’ of an uncertainty set exists for the Pred-DFL to be equivalent to Gen-DFL, finding it is intractable because ‘Pred-DFL + RO’ typically relies on fixed geometric shapes (e.g., ellipsoids, boxes) for the uncertainty set $\mathcal{U}$. If the true high-risk region of $p(c|x)$ does not match this geometry (e.g., it is multimodal or skewed), no single "size" of the ellipsoid can capture the risk without being overly conservative or unsafe.
>
> We thank the reviewer again for your comments and we hope our answers can address your concerns. We remain ready to address any further concerns if needed. Many thanks!

---

### Official Review · Reviewer_KwJ3 · 2025-10-31

**Soundness:** 2
**Presentation:** 2
**Contribution:** 2
**Rating:** 2
**Confidence:** 4

**Summary:**

This paper proposes Gen-DFL, a “Decision-Focused Generative Learning” framework for robust decision-making under uncertainty. It extends decision-focused learning (DFL) by replacing deterministic point predictions with a conditional generative model (e.g., conditional normalizing flow) that captures the conditional distribution $p_\theta(c|x)$ of optimization parameters. The framework combines this with a CVaR-based risk-sensitive optimization objective, forming a generate-then-optimize (GTO) paradigm. Theoretical analysis provides regret bounds suggesting that Gen-DFL performs better than traditional DFL when variance, dimensionality, or nonlinearity increases.

**Strengths:**

The topic is timely and relevant, as robust decision-making under uncertainty is a rapidly growing area at the intersection of optimization, learning, and generative modeling. The proposed framework is general and conceptually interesting—it integrates generative modeling and decision-focused optimization in a unified formulation that, in principle, could be applied across a wide range of uncertain decision-making problems. The problem setup is clearly motivated, and the generate–then–optimize structure is intuitively appealing.

**Weaknesses:**

- Limited conceptual novelty: The central idea—modeling parameter uncertainty via a generative model and optimizing CVaR—is highly similar to existing work in distributionally robust DFL and end-to-end conditional robust optimization (E2E-CRO). The paper does not clearly articulate how Gen-DFL differs from or improves upon these prior methods.
- Mathematical presentation is confusing: As noted in the comments, notation is inconsistent ($p_\theta(c|x)$ vs. $q(c|x)$ in equation (7)); $w^\star$ is used even though it is inaccessible; and several definitions (e.g., for CVaR-based regret) are unclear. Some derivations are incomplete or informal, lacking well-defined assumptions and proof steps; and several mathematical objects (e.g., $R(x)$ in Theorem 5.4) are never defined.
- Lack of explanation for the surrogate loss function: A central component of the algorithm—the surrogate loss function used for training—is introduced without any explanation, justification, or derivation. Since the entire algorithm builds upon this surrogate, the omission makes the methodology difficult to follow and gives an impression of a lack of transparency, as if the authors are glossing over key technical details. This severely hurts the paper’s clarity and credibility.

**Questions:**

- What is the relationship of $p_\theta(c|x)$ and $q(c|x)$? Are they refering to the same thing?
- Since $w^\star$ is not observable in real-world decision-making, how is regret approximated during training and evaluation?
- The regret definition, why is CVar positioned outside the regret difference rather than inside in equation (5)? Note that these two quantities can be different

---

> ### Author Response · Authors · 2025-11-21
>
> We thank the reviewer for the insightful feedback. We think there are some misunderstandings regarding our core contributions, and we’d like to address your questions as below:
>
> **1. The central idea… is highly similar to existing work…**
>
> One of the core arguments in our manuscript is that the conventional robust optimization approaches are overly conservative and protect only against the worst-case outcome. While prior approaches [1, 2, 3] typically construct uncertainty sets or calibrated intervals to deliver coverage/robustness guarantees, Gen-DFL instead learns a full conditional generative model and optimizes it jointly with a CVaR-based decision objective so that **we don't just protect the most extreme worst-case in an uncertainty set**. This enables sampling from structured, potentially multi-modal distributions without assuming fixed uncertainty sets, which empirically yields stronger performance in high-dimensional and tail-risk-sensitive settings. Our core contribution is the introduction of a more nuanced risk-sensitive optimization paradigm in which our method can optimize the expected performance over the worst-$\alpha\%$ of all the scenarios rather than just the absolute worst-case.
>
> **2. Mathematical notations**
>
> First, regarding $p_\theta(c|x)$ and $q(c|x)$, they refer to two different things. $p_\theta(c|x)$ is the distribution that the model is trying to learn in order to do downstream optimization, while $q(c|x)$ is the approximation of the true $p(c|x)$. The main point of DFL is that one has to learn a $p_\theta(c|x)$ that focuses on the tail end of the distribution to help downstream risk-sensitive optimization. On the other hand, $q(c|x)$ prioritizse the "bulk" (mean/mode) of the distribution where probability mass is high, which is not helpful for risk-sensitive optimization as the decision quality is driven exclusively by the tails. (see our experiments in Figure 3, 5)
>
> Second, one does not need to know the ground-truth $w^\star$ during training, as we can just compute the first term of the regret function since the gradient of the second would be zero as it is independent of model parameters $\theta$.
>
> **3. Regret definition**
>
> Could you elaborate on which two quantities are different? CVaR is positioned outside the objective function $f(\cdot)$ in both equation (5) and the definition in section 4.2.
>
> **4. Lack of explanation for the surrogate loss function.**
>
> The surrogate function is well-studied in the DFL literature [4] and is used by most of the works in this line of research as the original loss function is not differentiable. We omitted the derivation here because it is not the core contribution of our framework and it is already well-studied. We are happy to add an additional appendix section to include its technical details for completion. We also want to emphasize that the main goal of the experiments is the decision quality under risk during inference, which is not evaluated using the surrogate as we don’t need to worry about differentiation during inference. As long as the training process leads to good decision quality under risk during inference, then we argue that the surrogate loss function serves its purpose.
>
> We hope our answers address your questions. If all your concerns are cleared, we’d hope that you could reconsider your assessment. Thanks.
>
> References:
>
> [1] Wang et al., "Learning Decision-Focused Uncertainty Sets in Robust Optimization"
>
> [2] Chenreddy and Delage, "End-to-end Conditional Robust Optimization"
>
> [3] Yeh et al., "End-to-End Conformal Calibration for Optimization Under Uncertainty"
>
> [4] Mulamba. Contrastive losses and solution caching for predict-and-optimize. (2025)

---

> ### Author Response · Authors · 2025-11-26
> **Follow-up**
>
> Dear Reviewer,
>
> as the discussion period is nearing its end, we wanted to ask if there are any follow-up points we can clarify.
>
> We believe we have responded to all questions and concerns raised. If there are no further points of clarification regarding the manuscript, we kindly ask that reviewer KwJ3 consider increasing their score to reflect the improvements and clarifications we have provided. We are happy to continue to engage in discussion and answer any questions!

---

### Official Review · Reviewer_DNJQ · 2025-11-01

**Soundness:** 2
**Presentation:** 2
**Contribution:** 2
**Rating:** 2
**Confidence:** 4

**Summary:**

This paper proposes GEN-DFL, a decision-focused learning approach for contextual stochastic optimization problems that uses generative models (e.g. normalizing flows) to generate samples that are fed into sample average approximation model used to optimize a CVAR objective. In particular, in analogy to classical DFL approaches, they propose to train the generative (prediction) model using a surrogate decision-oriented loss function in order to obtain “decision-focused” samples that minimize a CVAR regret. The paper proposes a contrastive surrogate loss function for this setting, and it provides a theoretical analysis that provides an error bound for the surrogate loss function and a quantification of the improvement of the regret obtained by their model in contrast to classical (single-point-forecast-based) DFL approaches.
In a set of computational experiments with classical predict-then-optimize, DFL, and a robust DFL approach as baselines, it is shown that the proposed approach yields much better results in terms of a CVAR-based regret.

**Strengths:**

- The paper addresses a critical weakness of classical decision-focused learning approaches, namely the fact that the optimization models rely on a single point forecast, and addresses that weakness by using a SAA-based approach to compute a CVAR objective.

- The paper provides a surrogate loss function that is then usable in a classical end-to-end learning pipeline, and quantifies the error introduced by the surrogate loss.

**Weaknesses:**

- The paper ignores the fact that there exist many papers that deal with contextual stochastic optimization, see e.g. the survey paper Sadaba et. al (2025) (A survey of contextual optimization methods for decision-making under uncertainty, European Journal of Operational Research, https://doi.org/10.1016/j.ejor.2024.03.020), many of them using SAA-based approaches that rely on generating samples from conditional distributions that are generated by machine learning models.

- Given those approaches, for me, the big (and obvious) question is: What is the benefit of training the (probabilistic) machine learning models used for generating the sample with a decision-oriented loss function instead of using classical loss functions? In other words: What is the benefit of step 2 (Model learning) in section 4.2 of the paper? This question is not answered in the computational experiments, and this is, in my opinion, the big weakness of the paper. If there is not a big benefit, why should the reader bother with the expensive DFL pipeline? Personally, I dot not think that the task-specific loss will bring a lot of advantage, but I would love to see results from experiments that show the contrary. Also, I would like to emphasize that in classical DFL, it is common to compare “predict-then-optimize” (with classically trained prediction models) to “predict-and-optimize” aka DFL. This is why I find it strange not to compare “generate-then-optimize” (for which there is a lot of literature, see section 4 of the abovementioned survey) to “generate-and-optimize” (which is obtained by using a decision-oriented loss function).

- In my opinion, the experiments are also flawed in a different way as the approaches use different objective functions. This similar to a setting in which one solves a classical stochastic optimization problem with an expected value objective, and another with a CVAR-objective, and then compares their performance in terms of CVAR – it would come to no surprise that the solution of the first perfroms much worse.

- The paper does not mention approaches such as the one Nathan Kallus, Xiaojie Mao (2022) Stochastic Optimization Forests. Management Science 69(4):1975-1994., which also uses a decision-focused approach to solve risk-averse objective functions involving CVAR, also using a sample-based approxiation objectives (this would also be a good benchmark approach)


In its present state, I find that the weaknesses sketched abive namely neglecting whole stream of relevant literature on contextual stochastic optimization and  not comparing  the decision-focused training of the generative model to a classically trained generative model using the same SAA-based CVAR loss function. If this decision-focused training is not significantly better, the main contributions of this paper are not significant enough to justify a publication at ICLR.

**Questions:**

- The most important question is: In your experiments, given a fixed number of samples, how does generate-then-optimizes using a generation model trained in a classical way compare to your approach that uses a surrogate decision loss for training

---

> ### Author Response · Authors · 2025-11-21
>
> We thank the reviewer for the insightful feedback. We will address your concerns regarding our manuscript in the following:
>
> **1. Comparison with Data-Driven Contextual Stochastic Optimization**
>
> First, we respectfully clarify that our experiments included comparison with data-driven contextual stochastic optimization approaches involving SAA and generated samples. As detailed in our manuscript (Fig. 7), our baselines include, LRO [2], E2E-CRO [3], and [4].
>
> [2] and [3] learn a parametric uncertainty set (e.g., an ellipsoid) that defines a hard boundary for the worst-case. Similarly, [2] learns a convex uncertainty set defined by a PICNN-based score function. Moreover, the methods in the referenced survey [1] predominantly rely on restrictive assumptions or simplified models for the conditional distribution. Specifically, residual-based methods [5] assume that the error distribution is independent of the covariates or rely on simple parametric scaling for heteroscedasticity. Weight-based methods [6] are non-parametric and generate discrete conditional distributions supported only on historical samples, limiting their ability to model complex, continuous manifolds or generate novel scenarios in high-dimensional settings. Furthermore, expected value-based models [6] reduce the entire distribution to a single point prediction, completely discarding information about tail risks and higher-order moments which are critical for risk-averse decision-making, which is one of the main points of our contribution (see our experiments from line 464 to 480). In contrast, GenDFL is more general and learns a full generative model that captures the entire conditional distribution of the uncertainty parameters without being confined to a specific geometry or other assumptions.
>
> The above approaches also operate within the conventional robust optimization paradigm, protecting against the absolute worst-case in the uncertainty set. GenDFL operates within a more nuanced risk-sensitive optimization paradigm and optimizes the expected performance over the worst-$\alpha $% of all the scenarios rather than just the absolute worst-case in a certain uncertainty set. As shown in Fig. 7, Gen-DFL consistently achieves lower regret compared to these baselines across different experimental settings.
>
> **2. Stochastic Optimization Forests (SOF)**
>
> Similarly, SOF is a non-parametric method that generates "weights" for historical samples based on tree splits derived from perturbation analysis and is therefore tailored for tabular data where tree-splitting is efficient, while Gen-DFL can make use of modern generative models to generate new samples from any continuous distribution.
>
> **3. Benefit of Decision-Focused Loss and the comparison against Generate-Then-Optimize**
>
> Classical generative training aims to minimize a global distance (e.g., KL divergence) between the learned and true distributions. This forces the model to prioritize the "bulk" (mean/mode) of the distribution where probability mass is high. However, for risk-averse optimization, the decision quality is driven exclusively by the tails. By training with the decision-focused loss, Gen-DFL implicitly prioritizes the regions of the sample space that influence the downstream CVaR, even if those regions have low probability mass.
>
> To your question: “I find it strange not to compare “generate-then-optimize…” In fact, our experiments included these parts: (1) In Figure 7, we compare against other data-driven contextual stochastic optimization methods and Gen-DFL is shown to be superior to the baselines, (2) in Figure 3, when $\beta=0$, the model is trained purely on the classical generative model loss rather than the decision loss, which is essentially the Generate-Then-Optimize baseline in your question. We apologize if our writing is ambiguous and we can put more emphasis on the essential parts of our experiments so that the readers won’t miss them in the revised manuscript.
>
> **4. Experimental Comparisons**
>
> We respectfully disagree that the comparison is flawed, but we apologize if the setup was ambiguous.
> First, all methods are evaluated on the same downstream optimization problem using the same objective function (CVaR regret).
> We are strictly comparing how different *training* paradigms prepare a model to solve this specific risk-averse problem. Comparing a model trained on expected value against a model trained on CVaR is not a flaw, but rather the demonstration of the *misalignment* inherent in standard methods when applied to risk-sensitive tasks.
>
> We hope our answers address your questions. If all your concerns are cleared, we’d hope that you could reconsider your assessment. Thanks.
>
> References:
>
> [1] Kallus, N., & Mao, X. (2023).
>
> [2] Wang, et al. (2025).
>
> [3] Chenreddy, M., & Delage, E. (2024).
>
> [4] Yeh, et al. (2024).
>
> [5] Deng Y., Sen S. (2022)
>
> [6] Bertsimas D., Kallus N. (2022)
>
> [7] Ferreira K.J., Lee B.H., Simchi-Levi D. (2016)

---

> > ### Comment · Reviewer_DNJQ · 2025-11-25
> > **Response to Authors' comment**
> >
> > 1. Thank you for your clarification. I see that you compare against other RO-based approaches, but still, the approaches that you refer to in the second paragraph of your response, do not operate with an RO-based worst-case framework, but with a classical (contextual) stochastic optimizatoin setting, where VaR or CVaR-based objective functions (or weighted combinations of expected value and VaR or CVaR) in a sample-or scenario-formulation are pretty standard. I agree with your response that your approach for generating the samples from the conditional distributions may be more sophisticated than others, but it would be interesting to see in how far this increase in sophistication is translated to better results. This would be much more relevant in my opinion than comparing to RO approaches that operate with a very different setting.
> >
> > 2. Again, I tend to follow your argumentation, but an experimental evaluation would help to make that point.
> >
> > 3. Thank you, I indeed did not clearly understand this, but your response addresses this point.
> >
> > 4. I understand your argumentation, but as sketched under 1., there are many well-known standard approaches from contextual stochastic optimimization where the objective function perfectly aligns with your problem settting (e.g. train a non-trivial probabilistic regression or generative model of your choice, use that to sample from the conditional distribution, feed those samples in an SAA model with CVAR objective) that are not considered in your benchmarking table. I am not against keeping your "misaligned" benchmarks in the comparison, but I am really missing those that are aligned.
> >
> > I am slightly increasing my score.

---

### Author Response · Authors · 2025-12-04
**Author Final Remarks by Authors**

We sincerely thank all reviewers for their constructive suggestions. Their feedback has helped us substantially improve the paper by clarifying our novelty, strengthening empirical support, and refining theoretical scope. We will incorporate new experiments and refined positioning to ensure the contribution is technically clear. We have also uploaded our revised manuscript addressing many of the concerns. To summarize:

**Clarification of Core Contribution** (See response to Reviewer DNJQ and KwJ3)

We clarified that Gen-DFL is not merely a follow-up to traditional data-driven Robust Optimization (RO). Instead, it introduces a nuanced risk-sensitive paradigm: whereas RO protects against the single absolute worst-case in a fixed set, Gen-DFL optimizes the expected performance over the worst-$\alpha$% of scenarios (CVaR).

**Regarding Theorem 5.4** (See response to Reviewer Uiro)

We addressed Reviewer Uiro’s specific theoretical question regarding the regret bound. We provided the derivation for the upper bound of the aggregated regret gap $\mathbb{E}[\Delta R(x)]$ (removing the absolute sign as suggested).

**Interpretation of Theorem 5.4** (See response to Reviewer eGUr)

We clarified that the purpose of the derived upper bound is error decomposition rather than proving universal dominance. The bound explicitly identifies the failure modes of baselines (Pred-DFL)—showing how their error explodes with high variance and dimensionality—while demonstrating that Gen-DFL’s error is bounded primarily by the distributional distance (Wasserstein metric), effectively "shielding" it from variance-induced errors.

**Comprehensive Baseline Comparisons** (See response to Reviewer DNJQ)

We confirmed that our experiments do benchmark against state-of-the-art data-driven contextual stochastic optimization methods (including LR, E2E-CRO, and E2E-Conformal). We further distinguished Gen-DFL from Stochastic Optimization Forests (SOF), noting that while SOF is effective for tabular data via tree splits, Gen-DFL utilizes deep generative models to handle high-dimensional, continuous parameter spaces.

---

### Meta-Review · Area_Chair_FkvJ · 2025-12-27

**Summary:**

Existing predict-then-optimize (PTO) and decision-focused learning (which is predict-and-optimize) frameworks suffer from high-dimensional and risk-sensitive environments, since they rely on point estimates and fail to capture the full distributional uncertainty required for robust decision-making.  This paper presents Gen-DFL, a decision-focused learning for contextual robust optimization, which can be viewed as a generate-and-optimize framework where the model is  trained end-to-end using a loss function that combines CVaR-based decision regret and a generative regularization term.  The paper also provides a theoretical upper bound on the regret gap between the proposed Gen-DFL method and the baseline Pred-DFL method, showing that Gen-DFL's advantage grows with problem dimensionality, parameter variance, and risk sensitivity. The proposed method seems to be sound and its pipeline aligns well with the current end-to-end learning for contextual robust optimization. These are a few critical concerns which are well resolved by the authors.
1. Regarding the comparison with data-driven contextual stochastic optimization, the authors well responded to it, saying Fig. 7 shows that Gen-DFL consistently achieves lower regret compared to a few baselines across different experimental settings.
2. Regarding the interpretation of the theoretical bound, Theorem 5.4 provides an upper-bound on the regret gap between the proposed Gen-DFL method and the baseline Pred-DFL method. It is true that the lower-bound would be a better measure of superiority, but I agree that the upper bound is a proper measure of risk-robustness.

On the other hand, there are other issues that should be clarified in the revision.
The proposed method is built on E2E framework with CVaR as a risk-averse measure. Both E2E framework and CVaR are known things. A core contribution might be using more sophisticated generative model which is trained in the E2E framework. Thus, it would be more important and interesting to see how far this sophistication (conditional normalizing flow) translates to better results than classical ones.  This would be much more relevant in Reviewer DNJQ's opinion than comparing to RO approaches that operate with a very different setting. This is one of issues pointed out by Reviewer DNJQ, which I agree with.

**Reviewer Concerns:**

The authors have done a nice job addressing most of the reviewers' concerns.

**Reviewer Scores:**

Two reviewers DNJQ and Uiro are expected to raise their scores.
It was claimed by the authors that Reviewer KwJ3's comment is most likely to be AI generated. Regardless whether it is AI generated or not, it is not informative review, so I did not take into account.

---

### Decision · Program_Chairs · 2026-01-26

Accept (Poster)